# Biomechanics of the Human Osteochondral Unit: A Systematic Review

**DOI:** 10.3390/ma17071698

**Published:** 2024-04-08

**Authors:** Matteo Berni, Gregorio Marchiori, Massimiliano Baleani, Gianluca Giavaresi, Nicola Francesco Lopomo

**Affiliations:** 1Laboratorio di Tecnologia Medica, IRCCS Istituto Ortopedico Rizzoli, Via di Barbiano 1/10, 40136 Bologna, Italy; matteo.berni@ior.it (M.B.); massimiliano.baleani@ior.it (M.B.); 2Scienze e Tecnologie Chirurgiche, IRCCS Istituto Ortopedico Rizzoli, Via di Barbiano 1/10, 40136 Bologna, Italy; gianluca.giavaresi@ior.it; 3Dipartimento di Design, Politecnico di Milano, 20158 Milano, Italy; nicola.lopomo@polimi.it

**Keywords:** human, osteochondral unit, articular cartilage, subchondral bone, trabecular bone, biomechanical analysis, mechanical behaviour, experimental approach, constitutive model, systematic review

## Abstract

The damping system ensured by the osteochondral (OC) unit is essential to deploy the forces generated within load-bearing joints during locomotion, allowing furthermore low-friction sliding motion between bone segments. The OC unit is a multi-layer structure including articular cartilage, as well as subchondral and trabecular bone. The interplay between the OC tissues is essential in maintaining the joint functionality; altered loading patterns can trigger biological processes that could lead to degenerative joint diseases like osteoarthritis. Currently, no effective treatments are available to avoid degeneration beyond tissues’ recovery capabilities. A thorough comprehension on the mechanical behaviour of the OC unit is essential to (i) soundly elucidate its overall response to intra-articular loads for developing diagnostic tools capable of detecting non-physiological strain levels, (ii) properly evaluate the efficacy of innovative treatments in restoring physiological strain levels, and (iii) optimize regenerative medicine approaches as potential and less-invasive alternatives to arthroplasty when irreversible damage has occurred. Therefore, the leading aim of this review was to provide an overview of the state-of-the-art—up to 2022—about the mechanical behaviour of the OC unit. A systematic search is performed, according to PRISMA standards, by focusing on studies that experimentally assess the human lower-limb joints’ OC tissues. A multi-criteria decision-making method is proposed to quantitatively evaluate eligible studies, in order to highlight only the insights retrieved through sound and robust approaches. This review revealed that studies on human lower limbs are focusing on the knee and articular cartilage, while hip and trabecular bone studies are declining, and the ankle and subchondral bone are poorly investigated. Compression and indentation are the most common experimental techniques studying the mechanical behaviour of the OC tissues, with indentation also being able to provide information at the micro- and nanoscales. While a certain comparability among studies was highlighted, none of the identified testing protocols are currently recognised as standard for any of the OC tissues. The fibril-network-reinforced poro-viscoelastic constitutive model has become common for describing the response of the articular cartilage, while the models describing the mechanical behaviour of mineralised tissues are usually simpler (i.e., linear elastic, elasto-plastic). Most advanced studies have tested and modelled multiple tissues of the same OC unit but have done so individually rather than through integrated approaches. Therefore, efforts should be made in simultaneously evaluating the comprehensive response of the OC unit to intra-articular loads and the interplay between the OC tissues. In this regard, a multidisciplinary approach combining complementary techniques, e.g., full-field imaging, mechanical testing, and computational approaches, should be implemented and validated. Furthermore, the next challenge entails transferring this assessment to a non-invasive approach, allowing its application in vivo, in order to increase its diagnostic and prognostic potential.

## 1. Introduction

The human osteochondral (OC) unit is a multilayer structure composed of hyaline cartilage, here referred to as articular cartilage (AC), subchondral bone (SB), and trabecular bone (TB) (Figure 1). This complex structure limits contact pressure within the joints of the musculoskeletal system and distributes the surface load to the underlying bone, while still allowing a sliding motion between bone segments [1]. While the AC covering the joint surface provides low-friction motion, each layer of the OC unit contributes to the overall mechanical stiffness of the joint surface and undergoes different levels of strain during locomotion [2,3,4].

Maintaining the joint functionality relies on the homeostasis of the OC tissues, i.e., the synergic mechanical crosstalk between cartilaginous and mineralised tissues [5,6]. Despite the fact that there is not yet a full understanding of the factors promoting or impairing the homeostasis—and, consequently, the mechanical response—of the OC tissues, evidence suggests that changes in tissue strain levels may alter biochemical signals among tissues, as supported by mechano-regulation theories [7,8]. Traumatic events and the onset of pathologies such as osteoarthritis (OA) (for detail about the burden of OA see Section A.1) can induce alterations in the strain distribution across the OC tissues, modifying the biochemical signals that regulate cellular functions [9] and, therefore, producing changes in the main features, i.e., structure and composition, of the OC tissues [10,11,12]. AC tissue degradation, characterised by the loss of superficial proteoglycans (PGs), degradation of collagen fibres, and an increase in interstitial fluid, leads to fibrillation of the AC surface, which is the first visible sign of the pathology onset [13,14]. A significant increase in the remodelling of the SB tissue has also been reported [15]. Over time, these changes degrade the mechanical behaviour of the OC unit, further exacerbating alterations in strain levels [16] and, finally, leading to changes beyond tissues’ recovery capabilities. A non-physiological mechanical environment leads chondrocytes to stimulate catabolic activity [17,18] with the consequent degeneration of the tissue via PG depletion, increased tissue permeability, and destruction of the collagen network [19,20]. AC delamination occurs, leading to joint space narrowing and, ultimately, surface denudation [21]. Microfractures and neovascularisation take place in the SB, triggering a reparative response with inflammation, leading, over time, to subchondral sclerosis, osteophytosis, and cystic lesions [22,23,24,25].

As no effective treatments are currently available, a comprehensive understanding of the OC unit’s biomechanics is essential to (i) elucidate its overall response to intra-articular loads and, most importantly, (ii) assess the efficacy of proposed treatments in restoring physiological levels of strain before irreversible damage occurs. Additionally, an in-depth knowledge of the mechanical behaviour of the whole OC unit is crucial for the development of tissue engineering approaches as an alternative to arthroplasty for severely altered OC units, with consequent impaired joint function.

An examination of the literature reveals a lack of recent reviews specifically addressing the OC unit, although there is a very recent review focusing on the human AC [26]—in which the authors conclude by stating that there are still open gaps in understanding the biomechanical properties of the tissue, and that future studies need to investigate them at different dimensional scales—and three others on the biomechanics and mechanobiology of the TB [27,28,29], suggesting that the study of the biomechanics of both healthy and pathological TB remains an important way of understanding tissue complexity.

This systematic review aimed to provide an overview of the current trends and evidence achieved through experimental techniques in studying the mechanical behaviour of the OC unit. Firstly, a brief paragraph is included to provide foundational knowledge on the constitutive models developed until 2000 for describing the mechanical behaviour of the OC tissues. Despite the introductory nature of this first section, the reported information is essential for a thorough understanding of the findings presented in this review. Secondly, the results of a systematic search are reported, with a specific focus on the studies that evaluated the mechanical behaviour of the OC tissues comprising the human lower-limb joints, i.e., hip, knee, and ankle, through experimental testing. Studies conducted between January 2000 and December 2022 were identified via a search on online electronic databases. Data from eligible studies are retrieved, i.e., by focusing on (i) the type of tissue, (ii) the experimental test, (iii) the dimensional scale, and (iv) the constitutive models used to describe the mechanical behaviour of the tissues. Through the implementation of a multi-criteria evaluation approach—which considered methodology, data processing, and constitutive models—the reliability and reproducibility of each study were assessed. Then, the current trends in the evaluation of the OC unit are reported, revealing (i) which lower-limb joints and OC tissues are the most investigated and (ii) the main constitutive models and parameters describing the mechanical behaviour of the OC tissues. Last, a brief paragraph is included, highlighting the main techniques that can be employed to perform a comprehensive assessment of the OC unit, e.g., to investigate the correlations between tissue structure or composition and its mechanical behaviour.

By establishing the current understanding and identifying the areas lacking in OC unit biomechanics knowledge, along with considering recent proposed approaches to address the gaps through advancements in imaging technologies and experimental techniques, it may be possible to elucidate the comprehensive response of the OC unit and to understand how degenerative pathologies impair its features. This knowledge could improve the diagnosis of joint pathologies and drive tissue engineering approaches towards the development of multi-layer scaffolds better mimicking the response of the OC unit.

## 2. Constitutive Models of the Last Century

The mechanical behaviour of biological tissues—defined as response to external mechanical stimuli—is strictly dependent on their structure and composition and represents a critical aspect determining their in vivo functions. Therefore, its understanding is essential to explain the standard physiology of the tissues, properly understand the onset and progression of pathologies, and, thus, pave the way for the improvement of treatments and tissue engineering approaches dealing with these kinds of diseases. The mechanical behaviour of tissues—including the osteochondral ones—is generally explained by using constitutive models describing the peculiar trend of the experimental data. Besides mathematical laws, such models are defined by boundary conditions—describing the environment surrounding the tissue, e.g., the bound between the AC and the underlying bone tissue, or some of the peculiarities of the testing technique—and theoretical assumptions—e.g., describing the tissue structure and composition. By considering the aim of this systematic review, in the following we reported the main constitutive models applied to study the mechanical behaviour of the OC tissues. A brief description of these models was provided in order to supply the basic knowledge required to better understand the findings retrieved from the systematic review. An exhaustive discussion about the peculiarities, advantages, and limitations of the constitutive models reported in this section can be found by following the references provided. 

Several constitutive continuum models have been proposed to describe the mechanical behaviour of AC, SB, and TB before failure, i.e., when subjected to normal physiological conditions. These models aim to describe the relationship between the applied load and the resulting tissue deformation at load levels the tissue can withstand without being damaged, i.e., no assumptions are made regarding unrecoverable deformation or failure of the tissue.

The simplest model is the linear elastic model. It assumes that the tissue deforms proportionally to the applied load. The knowledge of the elastic constants—the number of independent parameters depending on the material symmetry (isotropic/transversely isotropic/orthotropic/anisotropic)—allows the description of a perfectly linear mechanical response [30]. Nonlinear responses of the tissue can be predicted by using hyperelastic models [31]. These models assume that the tissue response is derivable from a strain energy density function. Different strain energy density functions have been used in the following models [32]: the neo-Hookean model [33]; the polynomial model [34]; the Fung model [35]; the Odgen model [36]; the Yeoh model [37]; the Arruda–Boyce model [38]; and the Gent model [39]. Whatever the definition of the function, tissue deformation depends on the applied load and does not depend on the loading rate. The aforementioned approaches have been used to describe the mechanical behaviour of the mineralised tissues of the OC unit [40] or the instantaneous response to indentation and the equilibrium response of the AC [41,42,43].

More complex models have been proposed to describe the mechanical response of both mineralised and cartilaginous tissues. Mineralised tissues are viscoelastic materials showing time-dependent mechanical properties [44,45,46,47]. The time-dependent response of the TB is only significant at high strain rates or under constant loads over time [45,48,49,50,51]. Therefore, either poroelastic models [52,53] or viscoelastic models [54,55] have been proposed to describe the time-dependent mechanical behaviour of the TB under such conditions. As models other than the two mentioned families have been used since the year 2000, the papers in which these models are described, and to which the reader must refer for details, are also cited here to provide a complete background [56,57,58,59,60]. On the other hand, the time-dependent response of the AC [61,62], due to tissue structure, cannot be neglected except for instantaneous and equilibrium conditions (see previous paragraph). Therefore, viscoelastic models have been proposed to describe the AC response over time [62,63]. However, a theory which separately considers the two phases of the AC soon became the focus of model development. Indeed, the isotropic biphasic model (in detail, an incompressible linearly elastic isotropic porous matrix filled with an incompressible nondissipative fluid) used to describe the mechanical response of the AC dates from 1980 [64]. This model is conceptually different from the poroelastic model [56], but the two models are equivalent under the assumption of incompressible constituents and quasistatic small deformations [65]. From an experimental point of view, they can predict the behaviour of the tissue in response to the stress–relaxation test [66,67]. The isotropic biphasic model has been evolved into a linear transversely isotropic biphasic model [68] to describe the anisotropic mechanical properties of AC. However, some inconsistencies between model predictions and the observed equilibrium and stress relaxation behaviour of AC have been reported [69].

In the isotropic biphasic model, the materials parameters are constant. The viscoelastic behaviour is determined by “the diffusional drag of relative motion of the interstitial fluid with respect to the solid matrix” [64]. Therefore, this model, as well as the mentioned linear transversely isotropic evolutions, overlooks the fact that the matrix shows an intrinsic viscoelasticity [70]. To describe the flow-independent viscoelasticity of the AC, the biphasic model has been modified, first by making the solid phase viscoelastic in shear and elastic in bulk deformation [71], and then by making the solid phase viscoelastic in both shear and compression [72].

None of these models take into account the negatively charged glycosaminoglycan chains, referred to as fixed charges. The interaction between fixed charges and mobile ions are responsible for mechano-electrochemical phenomena, i.e., the attraction of fluid into the tissue during swelling and the associated osmotic pressure [73,74]. The inclusion of an ion phase, specifically monovalent cations (Na^+^) has led to the development of the triphasic model [75]. This model has been further developed to include the fixed charges (quadriphasic model), i.e., by considering the porous solid to be electrically charged [76].

This brief description of the constitutive models proposed to describe the mechanical behaviour of the tissues as continua has intentionally neglected the damage criteria, which identify the limit working conditions of individual tissues, as the following part of this paper mainly focuses on the mechanical response of the tissues under non-critical loading conditions.

## 3. Methods

### 3.1. Eligibility Criteria

This review specifically focused on scientific studies that investigate the mechanical behaviour of the tissues composing the OC unit of human lower-limb joints—i.e., their response to mechanical stimuli—by experimental approaches, and those that describe them by the mechanical parameters proper of specific constitutive models fitting the experimental force–displacement (or stress–strain) data. The following criteria were used to include scientific studies: (i) tissues must be retrieved from human hip, knee, or ankle joints; (ii) experimental design must be aimed at determining the mechanical behaviour of the retrieved tissue, i.e., studies employing mechanical techniques and mechanical protocols to stimulate cells and/or tissue engineering constructs during culture were excluded; (iii) experimental data must be reported, i.e., computational studies relying on experimental findings retrieved from previous studies were not considered eligible; (iv) experimental data must not already be published in previous reports, i.e., studies reporting data previously published with the purpose of addressing a different research question were excluded; and (v) constitutive model/s must be used to compute the mechanical properties of tissues from experimental data.

### 3.2. Search Strategies

The present systematic search was performed on online electronic databases—PubMed, Scopus, and Web of Science Core Collection—according to the PRSIMA statement [77] (see Figure 2 for the PRISMA flow diagram; the PRISMA checklist is provided in the Appendix A). The search queries specific to each database are reported in the Appendix B (see Table A1 of Section A.2). Scientific studies from January 2000 to December 2022 were included in this review if they met the eligibility criteria (see Section 3.1). Abstracts, reviews, letters, comments to the editor, protocols and recommendations, editorials, and guidelines were excluded, together with scientific studies not written in English. 

### 3.3. Study Selection

The achieved lists of studies were firstly submitted to public reference managers—i.e., Mendeley, www.mendeley.com; Rayyan, rayyan.qcri.org—to eliminate duplicates. Studies were then screened by two reviewers (M.Be and G.M.), considering the title and abstract. Any disagreement in the screening process was discussed by all authors and the thought of the majority of the authors was considered as the decision-making choice. 

### 3.4. Extraction of the Data

Data from eligible studies were retrieved and tabulated according to the clustering reported in Table 1. Studies were clustered according to their type—i.e., modelling, methodological, investigative, or comparative—in order to properly evaluate the quality of their main features, which depend on such a type (see Section 3.5 and Appendix B—Table A2). In addition, the type of studies was also considered as a suitable indicator for suggesting which aspects should be addressed in the future by studies focusing on the OC unit’s mechanical behaviour. In this regard, the temporal trend of the eligible studies—clustered according to (i) the joint from which the tissues were retrieved and (ii) the evaluated tissue/s—could also provide information for indicating future research directions.

Therefore, the distribution over time of the eligible studies, classified according to the reported insight, was determined considering the publication year (Reference), by applying a moving average filter, with a sliding window of length 3, across neighbouring elements.

### 3.5. Systematic Assessment of the Scientific Quality of the Studies

With the purpose of quantitatively evaluating eligible studies, the adaptation of a specifically designed method [78] was proposed. To define the scientific quality of research studies, the use of a multi-criteria decision analysis method is requested [79]. The Best Worst Method (BWM) allows to perform such an analysis by few comparisons among the metrics defined, according to the decision-maker’s perspective [79]. By setting the weights of these metrics—computed according to a minmax model—and after summarizing their extent in a global information—Aggregated Quality—the BWM allows a direct comparison between eligible studies. Specifically focusing on the adaptation of the BWM to this systematic review, the following steps were carried out. First, the features (i) methodology, i.e., exhaustivity of the experimental approaches, (ii) data processing, and (iii) constitutive model/s (see Table 1) were assigned to a three-level rating, i.e., low, mid, and high (details reported in Appendix B—Table A2). The choice of focusing on such specific features lies in the fact that—according to the authors’ expertise—these aspects are the ones primarily defining the reliability of the studies focused on the evaluation of the OC tissues’ mechanical behaviour. Second, the global weights of the features were computed by defining numerical coefficients—with values related to the preference of the best and of the worst metrics over the others—and, consequently, by minimizing their maximum absolute difference [78]. Third, the ratings thus obtained were gathered in an aggregated score—Aggregate Quality Score, in the range between 0 (lowest performance) and 2 (highest performance)—according to the BWM. Details about the weights of the evaluated features and of the scoring system are reported in Appendix B (Section A.3 and Table A2). To highlight the most reliable experimental studies performed on the OC mechanical behaviour until now, the eligible studies with the highest methodology score were discussed, and their findings were tabulated. Therefore, information about (i) the testing techniques, (ii) the range of the parameters describing the mechanical behaviour of the tissue/s, and (iii) the impact of pathologies was discussed, differentiated by tissue, constitutive model—i.e., elastic, hyperelastic, viscoelastic, plastic, and dynamic—and by the investigated dimensional scale.

## 4. Results and Discussion

### 4.1. Selection of the Studies

The initial literature search recovered 5804 studies. Of those, 3169 studies were identified using Scopus, 1906 using Web of Science, and 729 were found on PubMed. Aiming to eliminate duplicates, articles were run through Mendeley and Rayyan, removing 1902 studies. The remaining articles—i.e., 3902—were then screened considering their title and abstract, resulting in 3699 studies discarded. The full text of the leftover 203 articles was then reviewed, allowing for the removal of 73 additional studies due to non-compliance with the eligibility criteria. The remaining studies—i.e., 130—were, finally, investigated. The overall process is detailed in Figure 2, which maps out the number of records identified, included, and excluded, as well as the reasons for exclusion. The review was registered to the OSF registers (doi:10.17605/OSF.IO/BDCXJ).

**Figure 2 materials-17-01698-f002:**
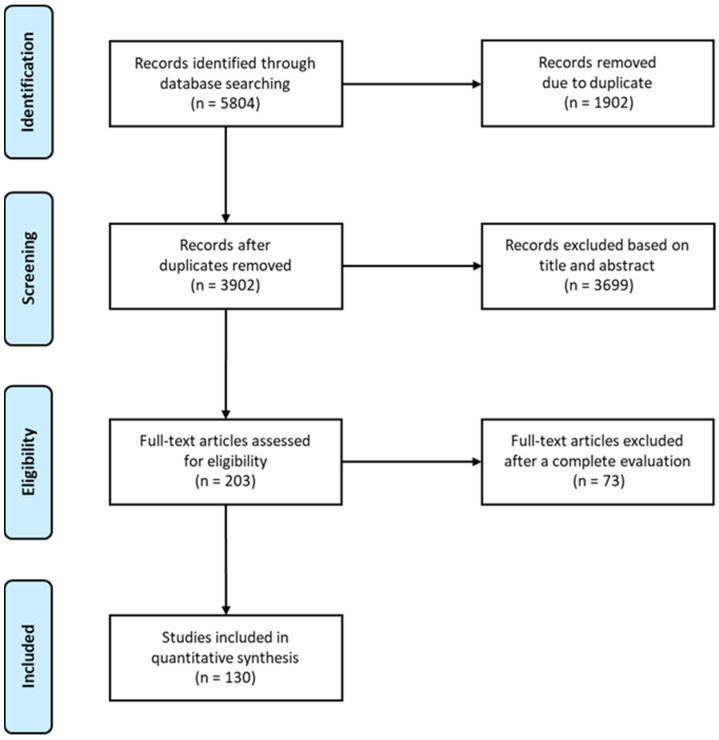
PRISMA flowchart for the eligibility of studies.

### 4.2. Trend over Time of Eligible Studies

The experimental studies investigating the mechanical behaviour of the OC tissues belonging to human lower-limb joints show an increase over time (“Total” series in Figure 3, which shows the temporal trend of studies divided by tissue or joint). By focusing on the investigated tissue, the majority of eligible studies assessed the mechanical response of AC (54%), followed by TB (42%), and SB (4%). An increase over time of the studies evaluating AC is suggested, while TB and SB studies seem to have reached an overall stable trend since 2016.

Concerning the joint from which tissues are retrieved, the knee seems to be the most common articulation (60%), followed by the hip (36%) and the ankle (5%). The number of studies focusing on knee OC tissues suggests an increase over time, while studies on hip and ankle joints show a decreasing and stable trend, respectively. 

According to these findings, the increase in studies focusing on the knee and on AC is the primary factor contributing to the suggested gradual rise in the mechanical assessment of OC tissues. Therefore, some mechanical evaluations are still missing, i.e., ankle tibial AC, hip acetabular SB, and ankle talar TB. Clearly, the most studied tissues and joints are the most harvestable ones, with a large articular surface, e.g., a lot of articular tissue can be harvested from joint arthroplasty, where replacements of hips and knees are far more common than of ankles; SB is thinner than AC and TB and, therefore, more difficult to be tested (see Section 4.4). This evidence does not imply that there are less important tissues or joints but, conversely, that there is a gap worth investigating. For example, the SB has been shown to be a possible target for preventive measures of OA [80], while, in relation to the ankle, total arthroplasty has recently gained interest [81]. Therefore, scientists should prove that results regarding specific tissues and sites can be generalised and should advance experimental techniques to reach more difficult—in terms of testability—tissues and articulations, eventually reaching the exploration of the whole OC unit (see Section 4.3, Section 4.4, Section 4.5 and Section 5).

Regarding the Aggregate Quality Score of the eligible studies, its temporal trend—differentiated by tissue and joint—is reported in Figure 4. Besides the few discontinuities highlighted by the studies focusing on the SB and the ankle, no visual differences among tissues and joints are suggested in terms of Aggregate Quality Score.

The temporal trend of the Aggregate Quality suggests an initial and progressive increase up to 2012–2013, from which a stable tendency seems to be reached. Starting from 2018 to 2019, the trend decreased instead, suggesting the implementation of less rigorous studies, i.e., overall of those investigating AC tissue and knee, more numerous (see Figure 3). By considering the range of the Aggregate Quality—i.e., between 0 and 2—and the scores obtained by the studies herein defined as eligible, the majority of them reached a score above the mid-level, i.e., 38% between 1 and 1.5 and 38% between 1.5 and 2, highlighting an overall good scientific quality of the experimental assessment of the OC tissues’ biomechanics. 

By looking at the Aggregate Quality Score of a specific study, it can be decided if more or less confidence should be given to the provided results. On the other hand, the scores can be used for selecting the most reliable and replicable studies of interest. A large amount of eligible studies (80 out of 130) did not reach the highest Aggregate Quality Score, e.g., because of lacking details about the testing protocol or missing/not suitable statistical analysis. Indeed, the scoring system herein proposed can also be useful as a guide considering the following fundamental, but still often missed, aspects: (i) experimental protocols should be designed and presented in a reliable and repeatable manner, allowing for a comparison among different studies; (ii) statistical analysis should be consistent to the specific design of the study, e.g., non-parametric methods should be preferred on parametric ones any time that assumptions on statistical distributions are not verified; (iii) a power analysis on sample size should be provided a priori or, at least, its absence should be properly discussed as a main limitation in arguing about the evidences achieved by the study.

As reported at the end of Section 3.5—and moreover considering that methodology is the feature primarily defining the Aggregate Quality of a study—in the following, the insights from eligible studies reaching the highest methodological score are reported, mainly differentiated by tissue, mechanical response, and investigated scale.

### 4.3. Articular Cartilage 

The mechanical behaviour of AC has been investigated primarily at the millimetre scale, with very few studies evaluating the response of the tissue at the microscale. The main mechanical features retrieved from studies focusing on the mechanical response of AC are related to its elastic behaviour (Table 2). In this regard, and despite AC response being described frequently as a linear elastic material, different assumptions and, thus, models are used to compute the relative mechanical properties of the tissue. The Hayes model is used to describe the tissue’s instantaneous elastic response, i.e., by defining the indentation of AC as a mixed boundary problem satisfying the field equations of the linear theory of elasticity for homogeneous, isotropic materials [41]. While the linear elasticity—and its relative parameters—can be derived from the analysis of experimental data obtained by different experimental techniques—i.e., tensile, shear, compressive, and indentation tests—the Hayes model is applied only to indentation-derived data.

Also regarding the evaluation of AC elastic behaviour, the biphasic [82,83] and the poro-viscoelastic fibril-network reinforced [84,85,86] models describe the response of the tissue more exhaustively, considering equilibrium phenomena. The biphasic model describes AC as a homogeneous binary mixture of an incompressible, isotropic, linearly elastic solid and an incompressible, inviscid fluid. In addition, such a model is based on the following assumptions, i.e., (i) the AC is a uniform layer of the biphasic material of thickness h, (ii) AC is bonded to the calcified-SB substrate, and (iii) the calcified-SB substrate is rigid and impervious to fluid flow. As the main outcome, the biphasic model provides the aggregate modulus of AC, representing the elastic behaviour of the tissue at equilibrium, i.e., once the viscosity related to both flow-dependent and independent phenomena are exhausted [69]. Despite mainly being included in the optimisation of the fitting procedure, additional parameters that can be retrieved by applying the biphasic model are Poisson’s ratio and the hydraulic permeability of the tissue [87]. The experimental tests providing boundary conditions suitable to verify the hypotheses of the biphasic model are indentation and compression, applied through a porous indenter or septum, respectively, and by creep or stress–relaxation protocols [88].

The poro-viscoelastic fibril-network reinforced model can be considered as a refinement of the previous model, in the effort of overcoming the main limitation related to the assumption that fluid-dependent phenomena are the only phenomena responsible for AC viscoelasticity. In this regard, such a model describes the contribution of the collagen network and of the non-fibrillar matrix—i.e., PGs saturated with fluid—separately, therefore considering their different role in withstanding the external load [88]. Moreover, the poro-viscoelastic fibril-network reinforced model defines the matrix as being supported by a network of nonlinear fibrils distributed in the radial, circular, and vertical directions. Consequently, the response to compressive stress is provided only by the non-fibrillar matrix, while the response to tensile and shear stress lies in the contributions of both fibrillar and non-fibrillar components. Through the application of such a constitutive model, it is possible to compute parameters related to the elastic response of AC, i.e., the fibril network, and the non-fibrillar matrix modulus, but also to calculate the permeability of the tissue. Moreover, the poro-viscoelastic fibril-network reinforced model allows the investigation of rate-dependent phenomena through testing protocols based on multi-step solicitations. The experimental tests employed to retrieve the parameters of such a model are compression and indentation—this last one is associated with a finite element approach [86,89], particularly by applying protocols based on stress–relaxation. 

According to the above reported findings, the elastic behaviour of AC at the millimetre scale has been primarily evaluated by indentation [89,90,91,92,93,94,95,96,97,98] and compression [92,98,99,100,101,102,103,104,105,106], followed by tensile [93,107] and shear [103] tests (Table 2). 

The combination of water and collagen contents accounted for approximately 25% of the variability in the compressive modulus [102]. Additionally, the compressive modulus depends on the rate of loading, which can induce a significant increase in its value [102]. Repetitive loading, such as that experienced during long-distance running, results in water exudation, as indicated by a decrease in T1rho relaxation times [100], positively correlated with water content, as measured using Magnetic Resonance Imaging (MRI) [108,109]. The local composition of AC determines its strain level, which has been demonstrated to be both depth-dependent and anatomical region-dependent [103].

Regarding the instantaneous modulus of AC, methodological studies suggest to properly define the boundary conditions of indentation tests, i.e., by defining experimental protocols based on nominal deformation and, moreover, considering the possible bias induced by different indenter diameters [90,98,107] (Table 2). In addition, the instantaneous modulus depends on the testing technique, providing higher values in the case of the tensile test, compared to confined and standard compression [107]. As reported for the elastic modulus, the instantaneous modulus also exhibits significant heterogeneity across articular surfaces [104]. With the purpose of providing methodologies suitable to investigate the AC’s mechanical response through routine clinical assessments, the outcomes achieved using the arthroscopic ultrasound method show results to be significantly correlated with the AC instantaneous modulus [96]. 

Besides instantaneous response, the equilibrium behaviour of AC—which can be computed using different constitutive models [86,110]—also strongly depends on the anatomical location [107], moreover being depth-dependent—at least considering the tissue tensile behaviour [93]. Regarding possible relations between AC structure/composition and its response, significant correlations were found (i) between the collagen orientation angle and the strain-dependent fibril network modulus and (ii) between the nonfibrillar matrix modulus and the PG content [111]. In the perspective of providing potential non-mechanical approaches evaluating the AC mechanical behaviour, quantitative MRI—i.e., through T1 and T2 relaxation times—can indirectly provide information on the tissue equilibrium modulus, as well as its site-dependent variations [105]. Moreover, optical spectroscopy based on the visible and near infrared range—retrieved using an arthroscopic approach—successfully predicted the AC equilibrium modulus [94]. 

**Table 2 materials-17-01698-t002:** Experimental studies focusing on AC elastic behaviour at the millimetre scale.

Reference	Type of Study	Pathology	Joint	Anatomical Position	Experimental Technique (Mode)	Dimensional Scale	Model	Stiffness (N/mm)	Elastic or Young’s Modulus, E (MPa)	Shear Modulus (MPa)	Electromechanical Quantitative Parameter	Poisson’s Ratio	Strain	Instantaneous Elastic Modulus E_0_, (MPa)	Strain-Dependent Instantaneous Elastic Modulus E_0_, (MPa)	Initial Fibril Network Modulus, E_f_ (MPa)	Strain-dependent Fibril Network Modulus, E_f_ (MPa)	Non-Fibrillar Matrix Modulus, E_m_ (MPa)	Equilibrium or Aggregate Modulus, E_eq_ or H_A_ (MPa)	Condition
Kurkijärvi et al. [105]	In	None	K	Femoral Condyle; Tibial Plateau; Patella; Trochlear Groove	Compression (stress–relaxation)	mm	LE	-	-	-	-	-	-	-	-	-	-	-	0.90 ± 0.43	No disease
Jeffrey et al. [98]	In	None	H	Femoral Head	Indentation; Compression	mm	LEI (Hayes)	-		-	-	-	-	10.3 ± 1.6 (indenter); 64 ± 13 (unconfined compression); 14.4 ± 3.5 (indenter, maximum modulus); 85.1 ± 4.9 (unconfined compression, maximum modulus)	-	-	-	-	-	No disease
Temple et al. [93]	In; Co	Aging	K	Femoral Condyle	Tensile test (stress–relaxation; dynamic mechanical analysis)	mm	LE	-	-	-	-	-	-	-		-		-	0.1–30.0 (Tensile, low strain rate); 0.1–70 (Tensile, high strain rate)	No disease
Keenan et al. [110]	In	None	K	Tibial Plateau	Indentation (creep)	mm	B (Mow)	-	-	-	-	0.00–0.05	-	-	-	-	-	-	0.48–1.58	No disease
Wong et al. [103]	In	None	K	Femoral Condyle; Tibial Plateau	Shear Test; Compression (stress–relaxation)	mm, µm (strain)	LE	-	0.1–0.9	0.01–5.00	-	-	0.01–0.40 (compressive); 0.01–0.50 (shear)	-	-	-	-	-	-	No disease
Deneweth et al. [104]	In	None	K	Tibial Plateau	Compression	mm	LE	-	-	-	-	-	-	7.0 ± 6.0 * (not covered by menisci); 10.0 ± 8.0 * (covered by menisci, anterior); 22.0 ± 15.0 * (covered by menisci, exterior); 20.0 ± 15.0 * (covered by menisci, posterior)	-	-	-	-	-	No disease
Griebel et al. [106]	In; Co	OA	K	Femoral Condyle; Tibial Plateau	Compression	mm	Anisotropic elasticity; depth dependent distribution of strain	-	-	-	-	-	0.0–0.12	-	-	-	-	-	-	No disease
Mäkelä et al. [111]	In; Co	OA	H	Femoral Head	Indentation (stress–relaxation)	mm	FRPVE	-	-	-	-	-	-	-	-	0.59 ± 0.48	0.61 ± 0.61	0.23 ± 0.22	-	No disease
Liukkonen et al. [96]	In; Me	None	K	Femoral Condyle	Indentation (stress–relaxation; dynamic mechanical analysis)	mm	LEI (Hayes)	-	-	-	-	-	-	0.1–0.4	-	-	-	-	-	No disease
Burgin et al. [102]	In	None	H	Femoral Head	Compression	mm	LE	-	1.1–3.3 (quasi-static); 0.5–4.98 (0.1 M Pa); 40–120 (impact)	-	-	-	-	-	-	-	-	-	-	No disease
Rautiainen et al. [95]	Co	OA	K	Tibial Plateau	Indentation (stress–relaxation; dynamic mechanical analysis)	mm	LE	-	-	-	-	-	-	-	-	-	-	-	1.2 ± 0.3	Early OA
Rautiainen et al. [95]	Co	OA	K	Tibial Plateau	Indentation (stress–relaxation; dynamic mechanical analysis)	mm	LE	-	-	-	-	-	-	-	-	-	-	-	0.2 ± 0.3	Advanced OA
Sim et al. [112]	In; Co	OA	K	Femoral Condyle; Trochlear Groove	Indentation; Compression (stress–relaxation)	mm	FRPVE	-	-	-	-		-	-		0.1–38 *		0.1–2.2 *	-	No disease
Afara et al. [94]	In	None	K	Femoral Condyle; Tibial Plateau; Trochlear Groove	Indentation (stress–relaxation; dynamic mechanical analysis)	mm	LE	-	-	-	-	-	-	-	-	-	-	-	0.9 ± 0.4 (0.15–2.14)	No disease
Waldstein et al. [20]	Co	OA	K	Femoral Condyle; Tibial Plateau; Patella; Trochlear Groove	Indentation (creep)	mm	B	-	1.0–17.0 *	-	-	-	-	-	-	-	-	-	0.4–2.4 *	OARSI grade 0
Waldstein et al. [20]	Co	OA	K	Femoral Condyle; Tibial Plateau; Patella; Trochlear Groove	Indentation (creep)	mm	B	-	1.5–8.0 *	-	-	-	-	-	-	-	-	-	0.3–1.5 *	OARSI grade 1
Waldstein et al. [20]	Co	OA	K	Femoral Condyle; Tibial Plateau; Patella; Trochlear Groove	Indentation (creep)	mm	B	-	0.5–9.5 *	-	-	-	-	-	-	-	-	-	0.2–1.3 *	OARSI grade 2
Waldstein et al. [20]	Co	OA	K	Femoral Condyle; Tibial Plateau; Patella; Trochlear Groove	Indentation (creep)	mm	B	-	1.0–7.5 *	-	-	-	-	-	-	-	-	-	0.3–1.4 *	OARSI grade 3
Waldstein et al. [20]	Co	OA	K	Femoral Condyle; Tibial Plateau; Patella; Trochlear Groove	Indentation (creep)	mm	B	-	1.0–4.5 *	-	-	-	-	-	-	-	-	-	0.3–1.2 *	OARSI grade 4
Waldstein et al. [20]	Co	OA	K	Femoral Condyle; Tibial Plateau; Patella; Trochlear Groove	Indentation (creep)	mm	B	-	1.0–2.0 *	-	-	-	-	-	-	-	-	-	0.2–1.0 *	OARSI grade 5
Nebelung et al. [99]	In	None	K	Femoral Condyle	Compression	mm	LE	-	0.419 ± 0.143	-	-	-	-	-	-	-	-	-	-	No disease
Sim et al. [92]	Co	OA	K	Femoral Condyle; Tibial Plateau	Indentation; Compression (stress–relaxation)	mm	LEI (Hayes); FRPVE	-	-	-	-	-	-	2.0 ± 1.0 *		8.5 ± 3.0 *	-	1.2 ± 0.1 *	-	Abnormal cartilage (ICRS grade > 0)
Sim et al. [92]	Co	OA	K	Femoral Condyle; Tibial Plateau	Indentation; Compression (stress–relaxation)	mm	LEI (Hayes); FRPVE	-	-	-	-	-	-	4.5 ± 1.0 *		13.0 ± 2.0 *	-	1.3 ± 0.2 *	-	Area surrounding abnormal cartilage
Sim et al. [92]	Co	OA	K	Femoral Condyle; Tibial Plateau	Indentation; Compression (stress–relaxation)	mm	LEI (Hayes); FRPVE	-	-	-	-	-	-	7.0 ± 1.0 *		18.5 ± 2.0 *	-	1.1 ± 0.2 *	-	Remaining normal articular cartilage (ICRS grade 0)
Sim et al. [97]	Co	OA	K	Femoral Condyle; Tibial Plateau; Patella	Indentation (electromechanical probe)	mm	LE	-	-	-	0.1 ± 0.5 *	-	-	-	-	-	-	-	-	ICRS grade 0
Sim et al. [97]	Co	OA	K	Femoral Condyle; Tibial Plateau; Patella	Indentation (electromechanical probe)	mm	LE	-	-	-	1.5 ± 0.6 *	-	-	-	-	-	-	-	-	ICRS grade 1
Sim et al. [97]	Co	OA	K	Femoral Condyle; Tibial Plateau; Patella	Indentation (electromechanical probe)	mm	LE	-	-	-	2.5 ± 0.6 *	-	-	-	-	-	-	-	-	ICRS grade 2
Sim et al. [97]	Co	OA	K	Femoral Condyle; Tibial Plateau; Patella	Indentation (electromechanical probe)	mm	LE	-	-	-	3.2 ± 0.5 *	-	-	-	-	-	-	-	-	ICRS grade 3
Sim et al. [97]	Co	OA	K	Femoral Condyle; Tibial Plateau; Patella	Indentation (electromechanical probe)	mm	LE	-	-	-	4.0 *	-	-	-	-	-	-	-	-	ICRS grade 4
Nebelung et al. [100]	In	OA (but macroscopically intact samples)	K	Femoral Condyle; Tibial Plateau	Compression	mm	LE	-	0.69 ± 0.40 (range, 0.20–1.69)	-	-	-	-	-	-	-	-	-	-	OA
Ebrahimi et al. [89]	Co	OA	K	Tibial Plateau	Indentation (stress–relaxation; dynamic mechanical analysis)	mm	LEI (Hayes); FRPVE	-	-	-	-	-	-	6.44 ± 4.85	56.09 ± 33.22	0.41 ± 0.37	15.42 ± 12.34	0.35 ± 0.28	1.19 ± 0.56	OARSI 0-1
Ebrahimi et al. [89]	Co	OA	K	Tibial Plateau	Indentation (stress–relaxation; dynamic mechanical analysis)	mm	LEI (Hayes); FRPVE	-	-	-	-	-	-	0.42 ± 1.34	50.05 ± 28.01	0.07 ± 0.17	18.29 ± 13.89	0.10 ± 0.05	0.42 ± 0.25	OARSI 2-3
Ebrahimi et al. [89]	Co	OA	K	Tibial Plateau	Indentation (stress–relaxation; dynamic mechanical analysis)	mm	LEI (Hayes); FRPVE	-	-	-	-	-	-	0.00 ± 0.76	21.68 ± 14.12	0.002 ± 0.07	7.65 ± 6.00	0.05 ± 0.04	0.21 ± 0.15	OARSI 4
Chokhandre et al. [107]	In; Me	None	K	Femoral Condyle; Tibial Plateau; Patella; Trochlear Groove	Tensile, Compression, and Confined compression (stress–relaxation)	mm	LE (Strain-dependent)	-	-	-	-	-	-	0.1–70.0 (Tensile); 0.1–8.0 (Confined Compression); 0.1–6.0 (Unconfined Compression)		-		-	0.1–60.0 (Tensile); 0.1–0.7 (Confined Compression); 0.1–0.8 (Unconfined Compression)	No disease
Ebrahimi et al. [91]	Co	OA	K	Tibial Plateau	Indentation (stress–relaxation; dynamic mechanical analysis)	mm	LEI (Hayes); FRPVE; Dynamic	-	-	-	-	-	-	0.1–12.0 *		0.01–0.9 *		0.15–0.80 *	0.65–2.1 *	OARSI 0-1
Ebrahimi et al. [91]	Co	OA	K	Tibial Plateau	Indentation (stress–relaxation; dynamic mechanical analysis)	mm	LEI (Hayes); FRPVE; Dynamic	-	-	-	-	-	-	0.1–3.0 *		0.01–0.35 *		0.10–0.20 *	0.20–0.80 *	OARSI 2-3
Ebrahimi et al. [91]	Co	OA	K	Tibial Plateau	Indentation (stress–relaxation; dynamic mechanical analysis)	mm	LEI (Hayes); FRPVE; Dynamic	-	-	-	-	-	-	0.1–2.0 *		0.01–0.10 *		0.01–0.15 *	0.10–0.50 *	OARSI 4
Berni et al. [90]	Me	None	K	Tibial Plateau	Indentation	mm	LEI (Hayes)	-	-	-	-	-	-	2.26–25.43	-	-	-	-	-	No disease

* = Data derived from graph. - = Data not reported. Type of study: In = Investigative. Co = Comparative. Me = Methodological. Pathology: OA = Osteoarthritis. Joint: H = Hip. K = Knee. Constitutive model: B = Biphasic. FRPVE = Fibril-reinforced poro-viscoelastic. LE = Linear elastic. LEI = Linear elastic isotropic.

The changes induced in AC response at the millimetre scale by pathological conditions were primarily investigated by considering the impact of OA (Table 2). In this regard, a significant effect of OA has been highlighted in terms of the AC instantaneous modulus [89,91,92]. In more detail, the AC initial and strain-dependent instantaneous modulus significantly decrease with the tissue degree of degeneration [89,91,92], e.g., by considering the Mankin [92] and the OARSI score [89,91]. These findings were ascribed mainly to changes in AC composition and structure, i.e., PG content and collagen orientation, both correlating with the initial instantaneous modulus [91]. By considering the perspective of retrieving arthroscopic findings about the AC mechanical response and composition, a quantitative parameter related to the elastic response and to the streaming potential of the tissue [112] is able to differentiate the degree of degeneration of AC [97]. 

The equilibrium response of AC is also strictly dependent on the pathological condition of the tissue, which significantly decreases with the OARSI score [20,89,91,95], suggesting that the alterations induced by OA are attributed to the deep relationships existing between the AC equilibrium modulus and both the structure and composition of the tissue, i.e., the PG content and collagen orientation angle [89,91].

The detrimental impact of OA is additionally highlighted by considering the parameters retrieved through the poro-viscoelastic fibril-network reinforced model, which separately describes the elastic behaviour of the collagen network and of the non-fibrillar matrix, comprising the tissue. In more detail, the initial fibril network modulus and non-fibrillar matrix modulus of AC are lower in early and advanced OA, compared to a non-pathological condition [89,91]. Moreover, significant relationships between such parameters and the instantaneous modulus of AC were also highlighted [92]. According to these findings, and also taking into account the heterogeneity of AC, the distribution of strains within the tissue—assessed using an in situ mechanical test, performed within an MR device—were demonstrated to be highly depth-dependent, moreover reflecting the severity of OA [106].

Through the employment of a complex constitutive model, such as the poro-viscoelastic fibril-network reinforced model, it is possible to compute not only parameters related to the elastic response of the different phases of AC, but also to the permeability of the tissue [110] (Table 3).

The permeability of AC is correlated with mechanical and electromechanical parameters describing the elastic response of the tissue [112] and is primarily investigated by studies evaluating the impact of OA [89,91,92,111,112]. In this regard, experimental studies highlighted that AC’s initial permeability is significantly related to the depth-dependent PG content, as well as to the collagen orientation angle [91]; furthermore, the coefficient describing the trend of strain-dependent permeability is correlated with both collagen orientation angle [91] and collagen content [111]. Being sensitive to both the structure and the composition of the tissue, it is not surprising that AC permeability is related to OA’s degree of degeneration [89,91,111].

Aiming to better describe the elastic response of AC to the extremely large deformation at which the tissue is subjected, hyperelastic constitutive models have been employed [113,114,115] (Table 4). In this regard, neo-Hookean, Yeoh [113,114], Veronda Westmann [114], and Gent [115] are the models proposed. The neo-Hookean and Yeoh models assume AC to be an isotropic and incompressible material, in which a strain energy function determines the relationship between shear modulus, principal strains, and the initial modulus of the tissue. Concerning the Verona Westmann model, its application assumes a uniaxial deformation of an isotropic and incompressible material [114]. This model describes the exponential dependence of stress on stretch by two independent parameters, i.e., a nonlinearity parameter and the shear modulus at zero strain [116]. The Gent model proposes a hyperelastic strain energy density function to describe the strain-stiffening phenomenon of soft materials [117]. Moreover, the Gent model can be further extended to interpret the molecular arrangement of collagen fibres in soft tissues [118].

**Table 3 materials-17-01698-t003:** Experimental studies investigating the AC poro-viscoelastic properties at the millimetre scale.

Reference	Type of Study	Pathology	Joint	Anatomical Position	Experimental Technique (mode)	Dimensional Scale	Model	Instantaneous Elastic Modulus E_0_, (MPa)	Strain-Dependent Instantaneous Elastic Modulus E_0_, (MPa)	Initial Fibril Network Modulus, E_f_ (MPa)	Strain-Dependent Fibril Network Modulus, E_f_ (MPa)	Non-Fibrillar Matrix Modulus, E_m_ (MPa)	Equilibrium or Aggregate Modulus, E_eq_ or H_A_ (MPa)	Initial Permeability, k (m^4^/N s)	Permeability Strain-Dependency Coefficient, M	Condition
Keenan et al. [110]	In	None	K	Tibial Plateau	Indentation (creep)	mm	B (Mow)	-	-	-	-	-	0.48–1.58	(1.7–5.4) × 10^−15^	-	No disease
Mäkelä et al. [111]	In; Co	OA	H	Femoral Head	Indentation (stress–relaxation)	mm	FRPVE	-	-	0.59 ± 0.48	0.61 ± 0.61	0.23 ± 0.22	-	(3.66 ± 2.86) × 10^−15^	17.26 ± 14.64	OA
Sim et al. [112]	In; Co	OA	K	Femoral Condyle; Trochlear Groove	Indentation; Compression (stress–relaxation)	mm	FRPVE	-	-	0.1–38	-	0.1–2.2	-	(0.0001–3) × 10^−12^	-	OA
Sim et al. [92]	Co	OA	K	Femoral Condyle; Tibial Plateau	Indentation; Compression (stress–relaxation)	mm	LEI (Hayes); FRPVE	2.0 ± 1.0 *	-	8.5 ± 3.0 *	-	1.2 ± 0.1 *	-	Trend across different regions	-	Abnormal cartilage (ICRS grade > 0)
Sim et al. [92]	Co	OA	K	Femoral Condyle; Tibial Plateau	Indentation; Compression (stress–relaxation)	mm	LEI (Hayes); FRPVE	4.5 ± 1.0 *	-	13.0 ± 2.0 *	-	1.3 ± 0.2 *	-	Trend across different regions	-	Area surrounding abnormal cartilage
Sim et al. [92]	Co	OA	K	Femoral Condyle; Tibial Plateau	Indentation; Compression (stress–relaxation)	mm	LEI (Hayes); FRPVE	7.0 ± 1.0 *	-	18.5 ± 2.0 *	-	1.1 ± 0.2 *	-	Trend across different regions	-	Remaining normal articular cartilage (ICRS grade 0)
Ebrahimi et al. [89]	Co	OA	K	Tibial Plateau	Indentation (stress–relaxation; dynamic mechanical analysis)	mm	LEI (Hayes); FRPVE	6.44 ± 4.85	56.09 ± 33.22	0.41 ± 0.37	15.42 ± 12.34	0.35 ± 0.28	1.19 ± 0.56	(1.19 ± 0.33) × 10^−15^	3.36 ± 2.07	OARSI 0-1
Ebrahimi et al. [89]	Co	OA	K	Tibial Plateau	Indentation (stress–relaxation; dynamic mechanical analysis)	mm	LEI (Hayes); FRPVE	0.42 ± 1.34	50.05 ± 28.01	0.07 ± 0.17	18.29 ± 13.89	0.10 ± 0.05	0.42 ± 0.25	(15.94 ± 47.45) × 10^−15^	4.19 ± 3.78	OARSI 2-3
Ebrahimi et al. [89]	Co	OA	K	Tibial Plateau	Indentation (stress–relaxation; dynamic mechanical analysis)	mm	LEI (Hayes); FRPVE	0.00 ± 0.76	21.68 ± 14.12	0.002 ± 0.07	7.65 ± 6.00	0.05 ± 0.04	0.21 ± 0.15	(20.88 ± 20.34) × 10^−15^	3.52 ± 4.45	OARSI 4
Ebrahimi et al. [91]	Co	OA	K	Tibial Plateau	Indentation (stress–relaxation; dynamic mechanical analysis)	mm	LEI (Hayes); FRPVE	0.1–12.0 *	-	0.01–0.9 *	-	0.15–0.80 *	0.65–2.1 *	Only significant correlations with the components of the tissue are reported	Only significant correlations with the components of the tissue are reported	OARSI 0-1
Ebrahimi et al. [91]	Co	OA	K	Tibial Plateau	Indentation (stress–relaxation; dynamic mechanical analysis)	mm	LEI (Hayes); FRPVE	0.1–3.0 *	-	0.01–0.35 *	-	0.10–0.20 *	0.20–0.80 *	Only significant correlations with the components of the tissue are reported	Only significant correlations with the components of the tissue are reported	OARSI 2-3
Ebrahimi et al. [91]	Co	OA	K	Tibial Plateau	Indentation (stress–relaxation; dynamic mechanical analysis)	mm	LEI (Hayes); FRPVE	0.1–2.0 *	-	0.01–0.10 *	-	0.01–0.15 *	0.10–0.50 *	Only significant correlations with the components of the tissue are reported	Only significant correlations with the components of the tissue are reported	OARSI 4

* = Data derived from graph. - = Data not reported. Type of study: In = Investigative. Co = Comparative. Pathology: OA = Osteoarthritis. Joint: H = Hip. K = Knee. Constitutive model: B = Biphasic. FRPVE = Fibril-reinforced poro-ciscoelastic. LEI = Linear elastic isotropic.

**Table 4 materials-17-01698-t004:** Experimental studies applying hyperelastic constitutive models to evaluate the AC mechanical behaviour at the millimetre scale.

Reference	Type of Study	Pathology	Joint	Anatomical Position	Experimental Technique (Mode)	Dimensional Scale	Model	Shear Modulus (MPa)	C10 Constant (MPa)	C20 Constant (MPa)	C1 Veronda Westmann (MPa)	C1 Veronda Westmann (a.u.)	Condition
Henak et al. [114]	In	None	H	Femoral Head; Acetabulum	Compression	mm	HE (neo-Hookean; Veronda Westmann)	5.32 ± 2.32	-	-	0.34 ± 0.24	6.55 ± 2.07	No disease
Robinson et al. [113]	Co	OA	K	Femoral Condyle; Tibial Plateau	Compression	mm	HE (neo-Hookean; Yeoh)	6.0 ± 1.6	1.7 ± 0.8	3.9 ± 3.4	-	-	No disease
Robinson et al. [113]	Co	OA	K	Femoral Condyle; Tibial Plateau	Compression	mm	HE (neo-Hookean; Yeoh)	4.6 ± 1.8	1.1 ± 0.8	2.0 ± 1.5	-	-	OA
Khajehsaeid et al. [115]	Co	OA	K	Femoral Condyle	Tensile	mm	HE (Gent)	Only normalised values were reported	-	-	-	-	OA

- = Data not reported. Type of study: In = Investigative. Co = Comparative. Pathology: OA = Osteoarthritis. Joint: H = Hip. K = Knee. Constitutive model: HE = Hyperelastic.

The main mechanical parameter provided by the above-reported models is the shear modulus [113,114,115], which can be retrieved by both compressive [113,114] and tensile tests [115]. Compared to the neo-Hookean model, the Veronda Westman model accurately predicted peak contact stress, average contact stress, contact area, and contact patterns developed by testing AC in compression [114]. Regarding how pathologies impair AC response, shear modulus, and both the Yeoh and Veronda Westmann models, constitutive parameters are significantly correlated with tissue structural parameters [113] and degeneration [113,115], endorsing the employment of such models to quantitatively assess the impact of OA on AC hyperelastic behaviour. 

The time-dependent response of AC has been investigated considering its highly viscous response, mainly related to both fluid-dependent and -independent phenomena (Table 5). In this regard, (i) the initial, transient response of AC is attributable to the collagen network re-arrangement, (ii) the time-dependent creep or stress–relaxation is primarily related to the interstitial fluid flow, and, finally, (iii) the equilibrium response is dependent on the properties of the extracellular matrix, ECM. With the purpose of assessing AC’s viscoelasticity, testing protocols based on creep are employed, therefore replicating the functioning of the tissue. Creep and creep rate are correlated with AC stiffness [119] and Young’s modulus [119,120]. In addition, the viscous strain induced within AC is linearly correlated with the Young modulus of the tissue; furthermore, it is hypothesised that a variation of such a strain could be linked to the permeability of the tissue [120].

Besides simple parameters [119,120], the viscoelastic behaviour of AC can be modelled by an isotropic viscoelastic constitutive law, which assumes that the components of the dilatation and deviatoric part of the stress tensor are decoupled [121]. In addition to the parameters related to the AC elastic behaviour, the viscosity coefficient of the tissue is significantly lower in pathological—i.e., affected by OA—than healthy conditions, moreover highlighting a relationship between such a coefficient and the grade of OA [121]. 

The assessment of AC plasticity can also provide information improving the knowledge about the degeneration of the tissue. In more detail, the strength of the tissue [93] and the energy of deformation [102] are the parameters investigated (Table 6).

Besides its relationship with the elastic modulus of the tissue, the energy of deformation—computed as the area underlying the loading curve, up to maximum compressive deformation—is correlated with the structural features of AC, i.e., thickness and water and collagen content [102]. Furthermore, the tensile strength of AC is site- and depth-dependent, highlight a significant decrease with age [93].

By considering the in vivo scenario at which AC is exposed, investigating the dynamic response of the tissue can represent a key aspect aiming to better understand not only its complex behaviour but, moreover, the impact of degenerative pathologies. The dynamic behaviour of AC was primarily investigated using compressive and indentation techniques, in particular applying testing protocols with a frequency falling in a range of (0.001 ÷ 88) Hz [89,91,122], even if the most used value is 1 Hz [94,95,96,105] (Table 7).

**Table 5 materials-17-01698-t005:** Experimental studies evaluating the time-dependent behaviour of AC at the millimetre scale.

Reference	Type of Study	Pathology	Joint	Anatomical Position	Experimental Technique (Mode)	Dimensional Scale	Model	Creep (mm)	Creep Rate	Viscosity Coefficient, η (MPas)	Condition
Barker et al. [120]	In	None	K	Femoral Condyle; Tibial Plateau	Indentation (stress–relaxation; dynamic mechanical analysis)	mm	VE	-	257–1352 *	-	No disease
Thambyah et al. [119]	In	None	k	Tibial Plateau	Indentation (creep)	mm	VE (model only for the elastic behaviour, i.e., LEI, Hayes)	0.05–0.23	-	-	No disease
Richard et al. [121]	Co; Mo	OA	H	Femoral Head	Indentation	mm	VE	-	-	218.7 ± 150.6	No disease
Richard et al. [121]	Co; Mo	OA	H	Femoral Head	Indentation	mm	VE	-	-	36.0 ± 41.4	OA

* = Data derived from graph. - = Data not reported. Type of study: In = Investigative. Co = Comparative. Mo = Modelling. Pathology: OA = Osteoarthritis. Joint: H = Hip. K = Knee. Constitutive model: VE = Viscoelastic.

**Table 6 materials-17-01698-t006:** Experimental studies evaluating the plasticity of AC at the millimetre scale.

Reference	Type of Study	Pathology	Joint	Anatomical Position	Experimental Technique (Mode)	Dimensional Scale	Model	Strength (MPa)	Energy of Deformation (mJ)	Condition
Temple et al. [93]	In; Co	Aging	K	Femoral Condyle	Tensile test (stress–relaxation; dynamic mechanical analysis)	mm	P	0.1–21 * (Tensile)	-	Alteration induced by the age
Burgin et al. [102]	In	None	H	Femoral Head	Compression	mm	P	-	75.5 ± 1.8	No disease

* = Data derived from graph. - = Data not reported. Type of study: In = Investigative. Co = Comparative. Joint: H = Hip. K = Knee. Constitutive model: P = Plastic.

**Table 7 materials-17-01698-t007:** Experimental studies evaluating the dynamic behaviour of AC at the millimetre scale.

Reference	Type of Study	Pathology	Joint	Anatomical Position	Experimental Technique (Mode)	Dimensional Scale	Model	Shear Storage Modulus, G′ (MPa)	Loss Modulus, G″ (MPa)	Dynamic Modulus, E_dyn_ (MPa)	Condition
Kurkijärvi et al. [105]	In	None	K	Femoral Condyle; Tibial Plateau; Patella; Trochlear Groove	Compression (stress–relaxation)	mm	D	-	-	7.83 ± 3.59	No disease
Liukkonen et al. [96]	In; Me	None	K	Femoral Condyle	Indentation (stress–relaxation; dynamic mechanical analysis)	mm	D	-	-	0.1–10 *	No disease
Rautiainen et al. [95]	Co	OA	K	Tibial Plateau	Indentation (stress–relaxation; dynamic mechanical analysis)	mm	D	-	-	6.8 ± 1.7	Early OA
Rautiainen et al. [95]	Co	OA	K	Tibial Plateau	Indentation (stress–relaxation; dynamic mechanical analysis)	mm	D	-	-	1.9 ± 2.3	Advanced OA
Afara et al. [94]	In	None	K	Femoral Condyle; Tibial Plateau; Trochlear Groove	Indentation (stress–relaxation; dynamic mechanical analysis)	mm	D	-	-	8.0 ± 3.5 (0.80—15.13)	No disease
Temple et al. [122]	In	None	H	Femoral Head	Compression (Dynamic Mechanical Analysis)	mm	D	A = 2.5 ± 0.6 MPa and B = 50.1 ± 12.5 MPa	4.8 ± 1.0 (range, 3.0–7.2)	-	No disease
Ebrahimi et al. [89]	Co	OA	K	Tibial Plateau	Indentation (stress–relaxation; dynamic mechanical analysis)	mm	D	-	-	6.87 ± 2.57	OARSI 0-1
Ebrahimi et al. [89]	Co	OA	K	Tibial Plateau	Indentation (stress–relaxation; dynamic mechanical analysis)	mm	D	-	-	3.69 ± 2.07	OARSI 2-3
Ebrahimi et al. [89]	Co	OA	K	Tibial Plateau	Indentation (stress–relaxation; dynamic mechanical analysis)	mm	D	-	-	1.67 ± 1.08	OARSI 4
Ebrahimi et al. [91]	Co	OA	K	Tibial Plateau	Indentation (stress–relaxation; dynamic mechanical analysis)	mm	D	-	-	Only the correlation coefficients with structure and composition of AC are reported	OARSI 0-1
Ebrahimi et al. [91]	Co	OA	K	Tibial Plateau	Indentation (stress–relaxation; dynamic mechanical analysis)	mm	D	-	-	Only the correlation coefficients with structure and composition of AC are reported	OARSI 2-3
Ebrahimi et al. [91]	Co	OA	K	Tibial Plateau	Indentation (stress–relaxation; dynamic mechanical analysis)	mm	D	-	-	Only the correlation coefficients with structure and composition of AC are reported	OARSI 4

* = Data derived from graph. - = Data not reported. Type of study: In = Investigative. Co = Comparative. Me = Methodological. Pathology: OA = Osteoarthritis. Joint: H = Hip. K = Knee. Constitutive model: D = Dynamic.

The dynamic modulus is the most often computed parameter, in particular by assuming AC to be an elastic and isotropic material [89,91,94,96]. The dynamic modulus of AC is correlated with the equilibrium modulus of the tissue [105] and, most importantly, with clinical findings [94,105]. In more detail, both quantitative MRI parameters—e.g., T1 and T2 relaxation times [95,105]—and the findings of optical spectroscopy—based on the visible and near infrared range [94]—are significantly related to the AC dynamic modulus.

The storage and loss modulus are additional parameters used to describe the dynamic behaviour of AC, in this case assumed to be a viscoelastic material [122]. By considering this approach, it is possible to elucidate the contribution of elasticity and viscosity to the comprehensive response of AC. In this regard, the storage modulus is at least double the loss modulus, suggesting that elasticity leads the response of the tissue. Moreover, the extent of both moduli depends on the frequency of the compressive stress [89,91,122].

Considering the possibility of using the dynamic behaviour as an indicator of AC structure and composition, the dynamic modulus is significantly correlated with the PG content and the collagen orientation angle [91]. As a consequence of such sensitivity, the dynamic modulus is significantly correlated with the OARSI score, i.e., being significantly reduced by early and advanced OA [89].

The mechanical response of AC was also evaluated at the micrometre scale, but only by very few studies [16,103,106] (Table 8). The micro-indentation technique is applied to investigate the viscoelastic and the dynamic behaviour of the tissue [16]. In this regard, the elastic, storage, and loss moduli of AC decreased significantly with the progression of OA; moreover, both the storage and loss moduli significantly decreased with age. Interestingly, a significant negative correlation was found between the AC storage modulus and the SB elastic modulus, supporting a deep link between such tissues.

Besides standard techniques, volumetric approaches are used to investigate—at a microscale level—the elastic behaviour of AC and the strain distribution within the tissue [103,106]. By using epi-fluorescence microscopy during bi-axial tests, it was highlighted that femoral AC is characterised by a higher elastic and shear modulus—and, consequently, it is subjected to lower axial and shear strain—compared to tibial AC, suggesting regional differences in the response of the tissue, even at the microscale level [103]. This finding was also highlighted by the compressive response of AC, investigated using a volumetric approach, employing MRI; moreover, such an approach allowed the authors to highlight a depth-dependent distribution of the compressive strain, with an extent that increases significantly through OA progression [106].

By looking at Table 2, Table 3, Table 4, Table 5, Table 6, Table 7 and Table 8, the range of mechanical parameters appears wide, even in the presence of similar, or even identical, mechanical testing modes, dimensional scales, and constitutive models (this finding can be also extended to SB and TB, as can be seen in the following paragraphs). This evidence can mainly be ascribed to the intrinsic tissue inhomogeneity and inherent variability of the samples. On the other hand, some of these differences may also arise from variations in the experimental procedures concerning tissue preparation and conditioning. Such differences can result in significant variations in the mechanical properties of the tissue, particularly when considering highly hydrated tissues like AC. Therefore, efforts to standardize testing protocols wherever possible are essential to carry out reliable studies that can serve as benchmarks, although the exploration of novel approaches is needed to address emerging research needs (e.g., strain rate changes, if representing quasi-static vs. dynamic vs. impact loading).

**Table 8 materials-17-01698-t008:** Experimental studies evaluating the mechanical behaviour of AC at the micrometre scale.

Reference	Type of Study	Pathology	Joint	Anatomical Position	Experimental Technique (Mode)	Dimensional Scale	Model	Elastic or Young Modulus, E (MPa)	Strain	Shear Storage Modulus, G′ (MPa)	Loss Modulus, G″ (MPa)	Condition
Wong et al. [103]	In	None	K	Femoral Condyle; Tibial Plateau	Shear Test; Compression (stress–relaxation)	mm, µm (strain)	LE	0.1–0.9	0.01–0.40 (compressive); 0.01–0.50 (shear)	-	-	No disease
Griebel et al. [106]	In; Co	OA	K	Femoral Condyle; Tibial Plateau	Compression	mm, µm	Anisotropic elasticity; depth-dependent distribution of strain	-	0.0–0.12	-	-	Different grades of OA severity
Peters et al. [16]	Co	OA; Aging	K	Femoral Condyle; Tibial Plateau	Indentation (Dynamic Mechanical Analysis)	µm	VE; D	0.04–8.13	-	0.90 ± 0.10 *	0.01–3.23 *	ICRS grade 0
Peters et al. [16]	Co	OA; Aging	K	Femoral Condyle; Tibial Plateau	Indentation (Dynamic Mechanical Analysis)	µm	VE; D	0.04–8.13	-	0.57 ± 0.07 *	0.01–3.23 *	ICRS grade 1
Peters et al. [16]	Co	OA; Aging	K	Femoral Condyle; Tibial Plateau	Indentation (Dynamic Mechanical Analysis)	µm	VE; D	0.04–8.13	-	0.27 ± 0.07 *	0.01–3.23 *	ICRS grade 2
Peters et al. [16]	Co	OA; Aging	K	Femoral Condyle; Tibial Plateau	Indentation (Dynamic Mechanical Analysis)	µm	VE; D	0.04–8.13	-	0.11 ± 0.05 *	0.01–3.23 *	ICRS grade 3
Peters et al. [16]	Co	OA; Aging	K	Femoral Condyle; Tibial Plateau	Indentation (Dynamic Mechanical Analysis)	µm	VE; D	0.04–8.13	-	0.16 ± 0.06 *	0.01–3.23 *	ICRS grade 4

* = Data derived from graph. - = Data not reported. Type of study: In = Investigative. Co = Comparative. Pathology: OA = Osteoarthritis. Joint: H = Hip. K = Knee. Constitutive model: D = Dynamic. LE = Linear elastic. VE = Viscoelastic.

### 4.4. Subchondral Bone 

The SB has received relatively less attention compared to the other OC tissues. This is likely because the SB forms a thin layer—up to 1.5 mm in thickness [123]—between AC and TB, preventing the extraction of thick specimens for mechanical assessments. In fact, the few studies focusing on the human SB in the last century have been performed on samples ranging from a few tenths of a millimetre to a millimetre in thickness [124,125,126]. It should be pointed out that the morphology of the tissue—characterised by a three-dimensional microchannel network—makes the results very sensitive to the porosity of the tissue, an aspect that is not taken into account. Therefore, more recent studies investigated the mechanical properties of the SB using micro- and nano-indentation tests [127]. 

The Oliver-Pharr method [128]—which assumes an elastic unloading response—was generally employed to determine the elastic modulus and hardness of the SB, considered to be a single-phase material [16,129,130]. The available data suggested a moderate correlation between the elastic modulus of SB and the age of the donor [16] (Table 9). Additionally, it has been found that the presence of pathologies such as OA alters the features of the SB, changing its mechanical properties; nevertheless, the relative trend does not appear to be monotonic. The remodelling process induced by early OA can affect the degree of mineralisation of the SB [23,131] (Table 9), which probably contributes to the observed dispersion in the collected data. In late OA, when morphological changes have already occurred and bone remodelling has slowed down [132], there appears to be an increase in the SB elastic modulus [16].

### 4.5. Trabecular Bone 

The TB has been extensively investigated across different dimensions, encompassing mainly the millimetre and micrometre levels. From a structural point of view, it was clear that the amount of TB per unit of volume—or apparent density—the intrinsic tissue properties, and the trabecular orientation impact on the apparent mechanical properties of the TB [133,134]. Therefore, the extraction of specimens with dimension in the range of (5 ÷ 10) mm must consider the principal structural directions of the tissue architecture, which can be described using a second rank tensor [135,136,137,138,139]. 

The TB is commonly modelled as a linear elastic material regardless of the scale, with a yield point identified using the 0.2% offset method [30]. The theoretical models describing the trabecular structure assume orthotropic symmetry for apparent elastic properties [140,141,142], although a transverse isotropy has also been reported for trabecular tissue retrieved from specific anatomical regions [143]. Hence, the apparent mechanical properties of the TB must be associated with the orientation of the structure in the direction of measurement.

By focusing on the elastic behaviour of TB at the millimetre scale—i.e., considering the tissue as a homogeneous material—compression is the experimental technique mainly applied [144,145,146,147,148,149,150,151,152,153,154,155] (Table 10). Tensile [146] and ultrasound [145] measurements are also proposed, the latter providing an estimate of the elastic modulus reasonably correlated to the one assessed using the compressive technique [145].

Regardless of the methods used to evaluate the TB, available data suggest the presence of correlations between the elastic modulus and the structural features of the tissue, i.e., with plate bone volume fraction, pBV/TV [153], axial bone volume fraction, aBV/TV [153], bone volume fraction, BV/TV [148,150,153], volumetric bone mineral density contributed by transverse trabeculae, tBMD [154], and only moderately with global density parameters [145,156]. Moreover, data reveal that apparent density—often expressed as bone volume fraction (BV/TV)—and tissue structural anisotropy account for up to 90% of the variation in experimentally measured elastic properties of healthy TB, modelled as a continuum [157], endorsing a high variability in tissue response. Differences across sites in on-axis modulus–density relationships may occur, suggesting that no single, universal prediction is achievable [146]. Considering TB directional behaviour, differences are determined by comparing directions parallel and non-parallel to the main orientation of the trabecular structure [144,147]. A great effect of the angle between the testing direction and the main direction of the bone structure on the compressive behaviour is highlighted, suggesting that the anisotropy exhibited by the microstructure of the TB reflects on its elastic response [158,159]. 

The composition and structure of TB undergo numerous changes with age and pathologies, both leading to a detrimental alteration of the tissue’s mechanical behaviour [160] (Table 10). No relation between the elastic modulus and the extent of deterioration has been found in the case of enzymatic and non-enzymatic glycation, despite these processes heterogeneously modifying the trabecular microarchitecture [151]. Regarding how diseases impact on the elastic response of the TB, type 2 diabetes (T2D) and hip fragility fracture—the latter adjusted for age, gender, and body mass index, BMI—impair the elastic modulus of the tissue more than OA and osteoporosis (OP) [152,155]. Patients affected by T2D have a particularly high risk of fracture, mainly due to the inferior microstructure and material properties of the bone tissue [161]. In addition, T2D patients exhibited low-grade inflammation—which negatively affects whole body metabolism and bone homeostasis [162]—altering bone cells’ activity and function [163].

Regarding the viscous response of the TB at the millimetre scale (Table 11), structural index and trabecular separation are the markers for evaluating the extent of deformation—applied using a compressive test with a protocol based on creep—at which the tissue is subjected [156]. Moreover, the creep rate of the TB depends on the tissue density, i.e., the higher the density, the lower the creep rate [156]. 

The plasticity of the TB at the millimetre scale is mainly investigated using compression and indentation tests (Table 12). Yield strength, stress and strain, and toughness are the parameters computed more often [144,148,149,150,151,152,153,154,155,161,164].

Trabecular plate-related parameters—i.e., plate and rod bone volume fraction, pBV/TV and rBV/TV; trabecular connection densities between plates, P–P Junc.D; average plate trabecular surface, pTb.S; and average plate trabecular thickness, pTb.Th—are significant predictors of variations in yield stress [153,154]. In addition, two- and three-dimensional measurements of TB structural features are strongly correlated with yield stress, endorsing the possibility of predicting its extent by findings retrieved from both planar and volumetric imaging [165]. Interestingly, relationships between apparent bone volume fraction, appBVF, and yield stress are also highlighted by considering findings retrieved using MRI, supporting the use of a clinical imaging technique not specific to mineralised tissues to determine differences in the mechanical response of TB [148]. The toughness of TB is also determined to be dependent on tissue structure—i.e., bone volume/total volume, BV/TV— highlighting a negative correlation with the amount of microdamage, e.g., linear microcracks [150]. Considering the directional behaviour of the TB, no differences are highlighted in terms of yield strength—i.e., by comparing directions parallel and non-parallel to the primary compressive orientation of the trabecular structure [144]—suggesting an isotropic plasticity of the tissue. Nevertheless, controversial findings—proposing the response of TB as anisotropic—are retrieved by taking into account the structure of the tissue—through the fabric tensor model—to compute yield properties and dissipated energy [149], the latter of which is strictly related to the maximal stress at which the TB is subjected [145].

As highlighted for elasticity, the plastic response of the TB may depend on the pathophysiological condition of the tissue. In this regard, T2D produces a more significant effect on the yield stress and toughness of the TB compared to OP and osteopenia [155]. Despite the fact that the toughness of the TB is not reduced in hip fragility fracture patients, smoking habits worsen the intrinsic trabecular mechanical performance, thus being associated with a lower stiffness and toughness [152]. Moreover, yield strain and toughness are also affected by enzymatic and non-enzymatic processes involved in TB turnover [151]. Lastly, the Brinell hardness of TB—i.e., hardness expressed as indentation depth—is not affected by hyperhomocysteinemia, which is a risk factor for OP [164]; this finding is probably due to the characteristics of the population—i.e., age and gender—from which the samples are retrieved.

**Table 10 materials-17-01698-t010:** Experimental studies and evaluating the elastic behaviour of TB at the millimetre scale.

Reference	Type of Study	Pathology	Joint	Anatomical Position	Experimental Technique (Mode)	Dimensional Scale	Model	Elastic Modulus (GPa)	Hardness (GPa)	Condition
Birnbaum et al. [144]	In	None	H	Femoral head	Compression	mm	LE	0.051–0.32	-	No disease
Pattijn et al. [145]	In; Me	None	H	Proximal femur	Ultrasonography (US); Compression	mm	LE	US: 0.052–0.306; UC: 0.021–1.514	-	No disease
Morgan et al. [146]	In	None	K	Proximal Tibia	Tension; Compression	mm	LE	0.1–3.0	-	No disease
Ohman et al. [147]	Me	None	H	Femoral head	Micro-indentation (Vickers); Compression (C)	µm; mm	LE	2.73 ± 1.06 (aligned); 1.59 ± 0.66 (misaligned)	32.5 ± 2.9 (aligned); 31.1 ± 3.1 (misaligned)	Aligned or misaligned to the trabecular main direction
Dall’Ara et al. [161]	In	None	H	Femoral head	Micro-indentation (Vickers); Compression	mm; µm	LE	0.5–4.5 *	32.9 ± 6.6 (wet); 35.1 ± 5.3 (dry); 44.6 ± 6.0 (embedded)	Wet vs. Dry vs. Embedded
Lancianese et al. [148]	In	None	K	Proximal Tibia	Compression	mm	LE	Discussed, without presenting computed values	-	No disease
Karim et al. [150]	In	None	K	Tibial plateau	Compression	mm	LE	Discussed, without presenting computed values	-	No disease
Schwiedrzik et al. [149]	In	None	H	Femoral head	Compression; Confined Compression	mm	LE	0.319 ± 0.164 (Compression)	-	No disease
Karim et al. [151]	In	NEG	K	Tibial plateau	Compression	mm	LE	Only coefficients of correlation with structural features are reported	-	No disease
Rodrigues et al. [152]	Co	OA	H	Femoral head	Compression	mm	LE	0.437 ± 0.237	-	OA
Rodrigues et al. [152]	Co	HF	H	Femoral head	Compression	mm	LE	0.324 ± 0.192	-	HF
Novitskaya et al. [156]	In; Co	OP	K	Proximal tibia	Compression	mm	LE	0.02–0.16 *	-	OP
Zhou et al. [153]	In	None	K	Proximal Tibia	Compression	mm	LE	0.27–1.58	-	No disease
Chen et al. [154]	In	None	A	Distal tibia	Compression	mm	LE	Only coefficients of correlation with structural features are reported	-	No disease
Yadav et al. [155]	Co	T2D	H	Femoral head	Nano-indentation (NI); Compression	µm; mm	LE	NI: 7 ± 2 *; C: 0.20 ± 0.10 *	0.25 ± 0.15 *	T2D
Yadav et al. [155]	Co	OP	H	Femoral head	Nano-indentation (NI); Compression	µm; mm	LE	NI: 9 ± 2 *; C: 0.35 ± 0.15 *	0.30 ± 0.25 *	OP
Yadav et al. [155]	Co	OPE	H	Femoral head	Nano-indentation (NI); Compression	µm; mm	LE	NI: 12 ± 2 *; C: 0.50 ± 0.20 *	0.75 ± 0.35 *	OPE

* = Data derived from graph. - = Data not reported. Type of study: In = Investigative. Co = Comparative. Me = Methodological. Pathology: HF = Hip fracture. NEG = Non-enzymatic glycation. OA = Osteoarthritis. OP = Osteoporosis. OPE = Osteopenia. T2D = Type 2 Diabetes. Joint: H = Hip. K = Knee. A = Ankle. Constitutive model: LE = Linear elastic.

**Table 11 materials-17-01698-t011:** Experimental studies evaluating the viscous behaviour of TB at the millimetre scale.

Reference	Type of Study	Pathology	Joint	Anatomical Position	Experimental Technique (Mode)	Dimensional Scale	Model	Final Creep Strain (µε)	Steady-State Creep Rate (sec^−1^)	Condition
Novitskaya et al. [156]	In; Co	OP	K	Proximal tibia	Compression	mm	VE	1600–6500 *	0.15–0.38 *	OP

* = Data derived from graph. - = Data not reported. Type of study: In = Investigative. Co = Comparative. Joint: K = Knee. Pathology: OP = Osteoporosis. Constitutive model: VE = Viscoelastic.

**Table 12 materials-17-01698-t012:** Experimental studies evaluating the plastic behaviour of TB at the millimetre scale.

Reference	Type of Study	Pathology	Joint	Anatomical Position	Experimental Technique (Mode)	Dimensional Scale	Model	Depth of Indentation at 1 kN (mm)	Yield strain (%)	Yield Stress/Strength (MPa)	Toughness (mJ/mm^3^)	Dissipated Energy Density (MPa)	Absorbed Energy at 20% Strain (J)	Condition
Birnbaum et al. [144]	In	None	H	Femoral head	Compression	mm	P	-	-	2.2–7.6	-	-	-	No disease
Pattijn et al. [145]	In; Me	None	H	Proximal femur	Ultrasonography (US); Compression	mm	P	-	-	-	-	-	0.0008–0.1372	No disease
Lancianese et al. [148]	In	None	K	Proximal Tibia	Compression	mm	P	-	-	0.1–11.0 *	-	-	-	No disease
Steines et al. [165]	In	None	H	Proximal femur	Compression	mm	P	-	-	0.1–18.0 *	-	-	-	No disease
Holstein et al. [164]	Co	HCY	H	Femoral head	Indentation (Brinell)	mm	P	0.7–1.5 *	-	-	-	-	-	HCY
Holstein et al. [164]	Co	Control	H	Femoral head	Indentation (Brinell)	mm	P	0.7–1.6 *	-	-	-	-	-	Control
Karim et al. [150]	In	None	K	Tibial plateau	Compression	mm	P	-	-	-	0.002–0.044	-	-	No disease
Schwiedrzik et al. [149]	In	None	H	Femoral head	Compression; Confined Compression	mm	P	-	0.0144 ± 0.0022	-	-	5.668 ± 4.416	-	No disease
Karim et al. [151]	In	NEG	K	Tibial plateau	Compression	mm	P	-	Only coefficients of correlation with structural features are reported	-	0.001–0.067.5	-	-	No disease
Rodrigues et al. [152]	Co	OA	H	Femoral head	Compression	mm	P	-	-	8.7 ± 4.8	0.19 ± 0.18	-	-	OA
Rodrigues et al. [152]	Co	HF	H	Femoral head	Compression	mm	P	-	-	6.8 ± 4.1	0.13 ± 0.11	-	-	HF
Zhou et al. [152]	In	None	K	Proximal Tibia	Compression	mm	P	-	0.52–0.83	1.12–8.92	-	-	-	No disease
Chen et al. [154]	In	None	A	Distal tibia	Compression	mm	P	-	-	Only coefficients of correlation with structural features are reported	-	-	-	No disease
Yadav et al. [155]	Co	T2D	H	Femoral head	Nano-indentation (NI); Compression	µm; mm	P	-	-	3.5 ± 1.5 *	0.065 ± 0.010 *	-	-	T2D
Yadav et al. [155]	Co	OP	H	Femoral head	Nano-indentation (NI); Compression	µm; mm	P	-	-	5.0 ± 2.5 *	0.115 ± 0.010 *	-	-	OP
Yadav et al. [155]	Co	OPE	H	Femoral head	Nano-indentation (NI); Compression	µm; mm	P	-	-	6.5 ± 2.0 *	0.185 ± 0.020 *	-	-	OPE

* = Data derived from graph. - = Data not reported. Type of study: In = Investigative. Co = Comparative. Me = Methodological. Pathology: HCY = Increased serum homocysteine. HF = Hip fracture. NEG = Non-enzymatic glycation. OA = Osteoarthritis. OP = Osteoporosis. OPE = Osteopenia. T2D = Type 2 Diabetes. Joint: H = Hip. K = Knee. A = Ankle. Constitutive model: P = Plastic.

The response of the TB is also investigated at the micrometre and sub-micrometre scale [16,147,155,161,166,167] (Table 13). In this regard, experimental approaches based on indentation, i.e., micro- and nano-indentation, propose the Oliver-Pharr [128] or the Vickers method to determine the elastic modulus [16,155] and hardness [16,147,161] of the tissue. Although the degree of alignment to the trabecular main direction does not significantly affect the hardness at the micrometre scale [147], the preparation of TB tissue strongly modifies the extent of such a parameter [161].

Other approaches investigating the mechanical response of the TB at the microscale employ Digital Image Correlation (DIC)—combined with macroscopic compressive [166] and tensile [167] tests—to determine the distribution of both the displacement and elastic modulus. A modified version of Wagner’s relationship—whose original version relates calcium content and the bone elastic modulus—was experimentally verified by employing compression and DIC, which inform a finite element model, predicting the response of the tissue [166]. In this regard, a higher accuracy of the elastic modulus prediction was achieved by using a modified Wagner’s relationship, i.e., relating voxel elastic modulus, microCT-derived hydroxyapatite density, bone organics density, and the organics volume fraction [166]. The response of the TB at the sub-micrometre scale can also be described using the staggered model—suggesting an arrangement of the mineral particles in agreement with the distribution of gaps in the collagen fibril. By applying such a model, it is possible to highlight the nonlinear dependence of the stress–strain of single trabeculae to the applied load, thus better describing the initial portion of the stress–strain curve once the tissue is subjected to monotonic loads [167]. 

By considering how the physiopathological condition impairs the tissue response at the micro- and sub-micrometre scale, aging and OA seem to have no significant effect on the TB elastic modulus [16]. Instead regarding hardness, its extent is significantly modified by the pathologies that alter bone metabolism and structure, i.e., hardness decreases more for patients with T2D and OP than with osteopenia, probably due to a higher decrease in trabecular thickness, Tb.Th; trabecular number, Tb. N; and structural model index, SMI [155]. 

**Table 13 materials-17-01698-t013:** Experimental studies distinguished evaluating the mechanical behaviour of the TB at the micrometre scale.

Reference	Type of Study	Pathology	Joint	Anatomical Position	Experimental Technique (Mode)	Dimensional Scale	Model	Elastic or Tangent Modulus (GPa)	Hardness (GPa)	Condition
Dall’Ara et al. [161]	In	None	H	Femoral head	Micro-indentation (Vickers)	µm	EP	-	32.9 ± 6.6 (wet); 35.1 ± 5.3 (dry); 44.6 ± 6.0 (embedded)	No disease
Ohman et al. [147]	Me	None	H	Femoral head	Micro-indentation (Vickers); Compression	µm; mm	EP	2.73 ± 1.06 (aligned); 1.59 ± 0.66 (misaligned)	32.5 ± 2.9 (aligned); 31.1 ± 3.1 (misaligned)	No disease
Marinozzi et al. [167]	In	None	H	Femoral head	Microtensile	µm	Response of the trabecular bone at the nanoscale, by considering the tissue as a composite	0.8–3.2 *	-	No disease
Cyganik et al. [166]	In	DHD; HD; FHN	H	Femoral head	FE coupled with compression on cubic samples	µm	Young’s modulus distributions assigned to the finite element models following modified Wagner et al.’s (Young’s modulus calcium content) relationship	Distribution and error made by estimating the elastic modulus through Wagner’s law	-	No disease
Peters et al. [16]	Co	OA; Aging	K	Femoral condyle; Tibial plate	Nano-indentation	nm; µm	EP	12.33 ± 0.50 *	0.11–1.05	ICRS grade 0
Peters et al. [16]	Co	OA; Aging	K	Femoral condyle; Tibial plate	Nano-indentation	nm; µm	EP	12.57 ± 0.60 *	0.11–1.05	ICRS grade 0
Peters et al. [16]	Co	OA; Aging	K	Femoral condyle; Tibial plate	Nano-indentation	nm; µm	EP	12.01 ± 0.70 *	0.11–1.05	ICRS grade 0
Peters et al. [16]	Co	OA; Aging	K	Femoral condyle; Tibial plate	Nano-indentation	nm; µm	EP	12.94 ± 0.80 *	0.11–1.05	ICRS grade 0
Peters et al. [16]	Co	OA; Aging	K	Femoral condyle; Tibial plate	Nano-indentation	nm; µm	EP	12.07 ± 1.00 *	0.11–1.05	ICRS grade 0
Yadav et al. [155]	Co	T2D	H	Femoral head	Nano-indentation (NI); Compression	µm; mm	EP	NI: 7 ± 2 *; C: 0.20 ± 0.10 *	0.25 ± 0.15 *	T2D
Yadav et al. [155]	Co	OP	H	Femoral head	Nano-indentation (NI); Compression	µm; mm	EP	NI: 9 ± 2 *; C: 0.35 ± 0.15 *	0.30 ± 0.25 *	OP
Yadav et al. [155]	Co	OPE	H	Femoral head	Nano-indentation (NI); Compression	µm; mm	EP	NI: 12 ± 2 *; C: 0.50 ± 0.20 *	0.75 ± 0.35 *	OPE

* = Data derived from graph. - = Data not reported. Type of study: In = Investigative. Co = Comparative. Me = Methodological. Pathology: DHD = Degenerative hip disease. FHN = Femoral head necrosis. HD = Hip dysplasia. OA = Osteoarthritis. OP = Osteoporosis. OPE = Osteopenia. T2D = Type 2 Diabetes. Joint: H = Hip. K = Knee. Constitutive model: EP = Elasto-plastic.

### 4.6. Final Considerations on OC Tissues 

Previous sections highlighted the importance of the biomechanical characterisations of the specific OC tissues for deepening the knowledge of relevant physiopathological conditions, as emerged from many of the reviewed studies on the topic. Therefore, advancing those characterisations has clear clinical implications in terms of diagnostic and treatment (e.g., regenerative medicine) potential. In this regard, this systematic review identified the main gap, i.e., no experimental studies tested the functional response of the human OC unit as a whole system. Indeed, this represent a major limitation, in particular when considering the interplay between the OC tissues. Therefore, future research should focus on the development of testing protocols able to stress the whole OC unit and, in the meantime, to investigate and to model the response of its constitutive tissues. It needs a multi-disciplinary approach, envisaging complementary techniques, such as imaging that could bring us closer to in vivo applications. The next paragraph summarizes the techniques currently employed to integrate the insight retrieved from the mechanical testing of individual OC tissues and, moreover, how they can be applied to the whole OC unit (for details about the techniques—e.g., advantages, limitations, suitability for different research or clinical scenarios, illustrative tables/figures, etc.—refer to the studies reported in the following section).

## 5. Complementary Approaches to Investigate the OC Unit’s Biomechanics

To properly elucidate the biomechanics of challenging tissues, such as those comprising the OC unit, mechanical assessments must be supported using other methods. Despite the present review basing its ratings on mechanical experiments, it is important to highlight, in brief, the main approaches investigating tissue composition and structure reported by the studies investigated here. Indeed, to complement the mechanical information with that about structure/composition is fundamental to explain tissue function.

Regarding AC, biochemical assessments and histology still represent the gold standard in evaluating not only the main features of the tissue, but also the efficacy of treatments [168]. Biochemical analyses focus on tissue composition, e.g., of synovial fluid to investigate the inflammatory environment of a joint, while histology focuses on tissue/cellular structure and morphology, usually by optical inspection of tissue slices. Image-based techniques have emerged as accurate and reliable methods to investigate AC. Optical approaches—i.e., stereo, polarised light, and scanning electron microscopy—allowed us to evaluate the presence of damage on the articular surfaces [98,169], together with collagen orientation angle and content [111]. The main drawback of these techniques is that they are strictly ex vivo, being performed on biopsies or resected tissues.

Considering both ex vivo assessments and clinical applications, X-ray imaging has been used—thanks to advances in contrast-enhanced approaches—to detect chondral lesions and to evaluate the severity of AC degeneration [170]. MRI permit to assess AC homeostasis, both for preclinical and clinical applications [171]. Delayed gadolinium-enhanced MRI (dGEMRIC), T2, and T1rho are the main sequences allowing to quantitatively evaluate AC, relating the extent of the signal to the tissue composition. dGEMRIC sequences permit an indirect measure of AC GAG content, while T2 and T1rho signals are directly related to free water, indirectly to collagen content and orientation, and inversely to PG/GAG content, respectively [101,172,173,174]. Such sequences represent a powerful tool not only to investigate AC composition, but also to detect the onset of degenerative pathologies, like OA. However, the insights that can be retrieved through their analysis strongly rely on the resolution used to obtain the three-dimensional information. Consequently, care must be taken in comparing the findings of different studies and, therefore, efforts should be made in order to define standard acquisition protocols. Finally, Fourier transform infrared [95,175] and Raman spectroscopy [176] showed a great capability in investigating changes in collagen, PG, and water content, as well as mineral distribution within AC.

Concerning the mineralised tissues of the OC unit—i.e., SB and TB—X-ray imaging plays a key role in evaluating their structure and composition. Quantitative experimental, preclinical, and clinical computed tomography techniques are employed. These techniques permit to measure the bone mineral density and, when the isotropic spatial resolution is better than 100 microns with the highest accuracy being achieved with the resolution of the order of one micron (Synchrotron imaging), parameters related to tissue structure—i.e., bone volume fraction (BV/TV), bone surface area to total volume ratio (BS/TV), trabecular thickness (Tb.Th), trabecular separation (Tb.Sp), trabecular number (Tb.N), and structure model index (SMI) [177,178,179,180]. Significant differences have been found in these parameters between the bone tissue of different anatomical regions [181] and physiopathological conditions [182,183].

Correlating tissues’ structure and composition to their mechanical response is crucial to comprehensively assess the biomechanics of the OC unit, specifically considering how degeneration and treatment affect the homeostasis of such a complex unit. From this perspective, image-based techniques such as Digital Image Correlation (DIC) and Digital Volume Correlation (DVC) can collect comprehensive information related to OC tissues’ biomechanics, by evaluating how the tissues’ structure responds to external loads [184,185,186,187,188,189]. For both techniques, the resolution of the displacement map and, therefore, the ability to account for tissue heterogeneity, depends on the imaging resolution and the subset/sub-volume size [187]. With particular reference to DVC, exhaustive measurement of the inner deformation induced within tissues are reported for mineralised tissues [188,190,191] and, more recently, extended to AC and its junction with bone tissue [4,184,189,192,193]. Concerning the assessment of AC using X-ray-based DVC, two main approaches are currently used. The first approach mainly employs Phase Contrast, while the second employs Synchrotron imaging; both these approach have been shown to be suitable for detecting the microstructural features of AC [4,189]. Nevertheless, the main drawback of the Phase Contrast approach is the exposure of AC to repeated and prolonged irradiation, leading to the possible degradation of the tissue [187,194]. The second approach entails the staining of soft tissues via radiopaque contrast agents, thus increasing the contrast in the image and resolving their main features, e.g., the local arrangement of chondrocyte lacunae [184,192]. However, exposing AC to contrast agents could present two main limitations [195], as follows: (i) the possible alteration of tissue morphology and mechanical response and (ii) the enhancement reached by the penetration of a contrast agent could not induce the high-contrast three-dimensional variation of grayscale values required by DVC algorithms, reducing—or even impairing—the potential of DVC techniques. According to this evidence, and with the perspective of applying DVC to assess the field of displacements and strains within the OC unit, the use of contrast agents and specific imaging solutions must be thoroughly evaluated. 

The most futuristic, intriguing challenge can be represented by developing clinical imaging in such a way to apply DVC to the OC unit as physiologically loaded and, therefore, for investigating its mechanical performance in vivo. In this way, the diagnostic potential would strongly increase, i.e., revealing the impact of pathologies on articular tissues during their functioning.

## 6. Conclusions and Future Perspectives

This review is mainly focused on the mechanical behaviour of OC tissues, with the aim of highlighting the main achievements retrieved from the relative literature. Different models have been proposed for the description of the mechanical behaviour of the OC tissues. Among them, the poro-viscoelastic fibril-network-reinforced constitutive model has become common for describing AC, while the models describing the mechanical behaviour of mineralised tissues are usually simpler (i.e., linear elastic, elasto-plastic). Most advanced studies have tested and modelled multiple tissues of the same OC unit, but individually rather than through integrated approaches. This approach allowed an in-depth characterisation of the investigated tissue, especially of AC and TB, in well-controlled experimental conditions. It also allowed the validation of different models proposed to describe the mechanical behaviour of OC tissues. Additionally, the available experimental techniques that can achieve submicron resolution, combined with the current computing power technology, already make the development of multiscale models of the individual tissue that predict the mechanical environment of cells possible. 

However, the OC unit is a multilayered structure composed of different tissues, each one influencing the response of the others and contributing to the overall mechanical response. Therefore, the validation of multiscale models of such a multilayer structure requires accurate experimental data describing the comprehensive mechanical behaviour of the OC unit. The resolution achievable with optical techniques is crucial to provide accurate information on the physiological volumetric deformation occurring within the OC unit and to detect small-scale changes due to pathological status that alter the mechanical environment of the cells.

Although this approach is still challenging and requires a multidisciplinary approach, involving mechanical, biochemical, computational, and imaging techniques, it could provide insights into how the mechanical environment regulates cell response and triggers signalling cascades, driving the development of scaffolds mimicking the mechanical response of the OC unit to external loads.

## Figures and Tables

**Figure 1 materials-17-01698-f001:**
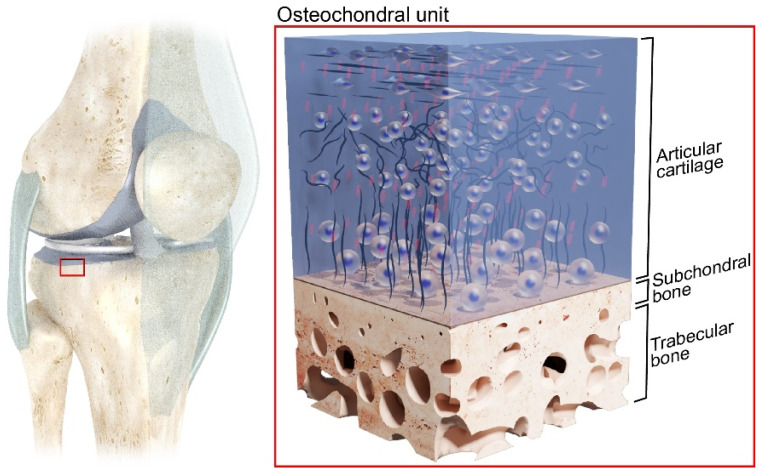
Scheme of the osteochondral unit structure, with particular focus on the knee joint, i.e., proximal tibia.

**Figure 3 materials-17-01698-f003:**
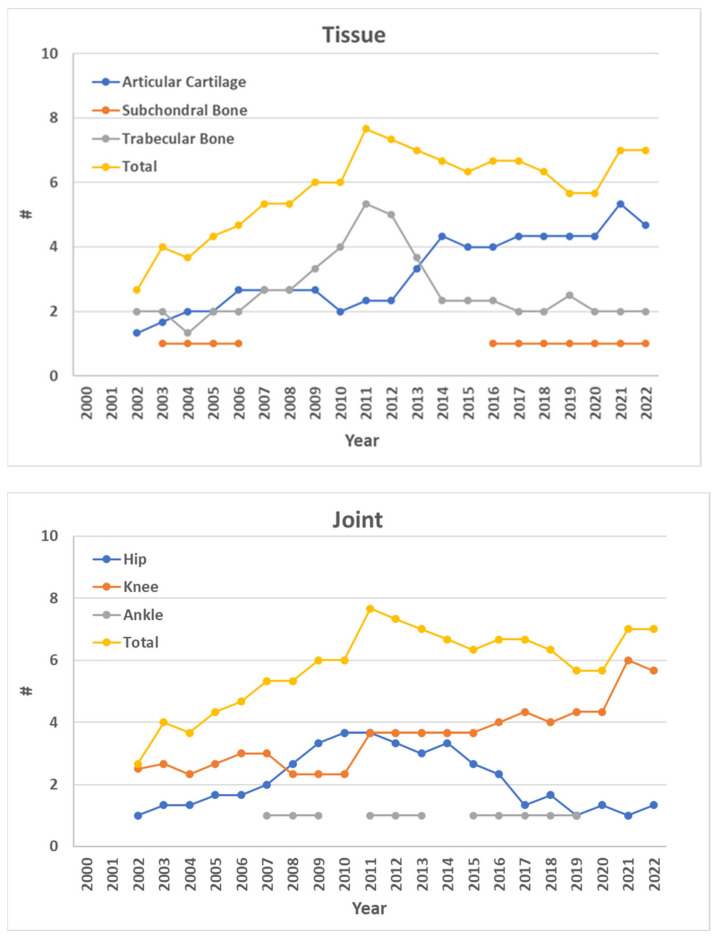
Distribution of eligible studies (total) over time. Studies are further clustered by considering the investigated tissue (**top graph**) and the joint from which samples are retrieved (**bottom graph**). A discontinuity in the line means that there are no eligible studies in a range of three years, centred on the year for which the point is missing.

**Figure 4 materials-17-01698-f004:**
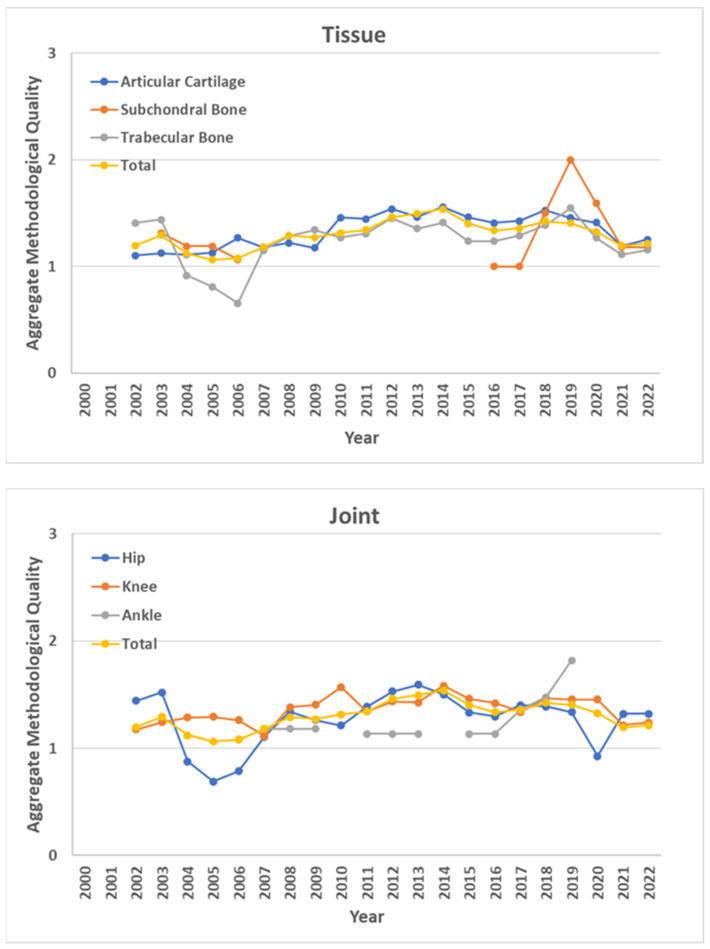
Distribution, over the years, of the Aggregate Quality Score of eligible studies, clustered by tissue (**top graph**) and joint (**bottom graph**). A discontinuity in the line means that there are no eligible studies in a range of three years, centred on the year for which the point is missing.

**Table 1 materials-17-01698-t001:** Insight retrieved from eligible studies.

Insight	Description
**Reference**	Title and publication year
**Joint**	Anatomical site/s—i.e., hip, knee, or ankle—from which tissue samples were excised
**Tissue**	Type/s of tissue/s—i.e., AC, SB, TB—evaluated
**Type of study**	Modelling, i.e., a study proposing a new constitutive model or updating an existing one, moreover providing an experimental application
	Methodological, i.e., a study proposing a new experimental approach, or evaluating how the tissues’ response varies depending on the testing parameters
	Investigative, i.e., a study evaluating the mechanical parameter/s of tissue/s by investigating distribution across the sample geometry, or a dependence on joint/s or on donor/s
	Comparative, i.e., a study investigating how a pathological disease, or a specific treatment, modifies the mechanical response of tissue/s compared to control/healthy condition
**Layers**	Whether a study focused on single tissues or evaluated the mechanical behaviour of the whole osteochondral unit. The latter option means that the mechanical behaviour of multiple layers was investigated and modelled simultaneously
**Mechanical test**	Type of experimental approach applied to investigate tissue behaviour, e.g., compression, indentation, tensile, and shear test
**Dimensional scale**	Scale—i.e., millimetric, mm; micrometric, um; and/or nanometric, nm—at which the mechanical behaviour of tissue/s was evaluated
**Constitutive model/s**	Mathematical model/s used to compute the mechanical parameters of tissue/s; moreover, type and numerosity of such parameters were also retrieved
**Mechanical properties**	Values of the mechanical parameters, strictly dependent on the constitutive model used to fit the experimental data, e.g., Young’s modulus (elastic), creep (viscous), shear storage modulus (dynamic), dissipated energy (plastic behaviour). Moreover, the numerosity of the computed parameters was also noted.
**Data processing**	Highlights presence and reliability of statistical data analysis. Meaning of the analysis depends on the type of study, as follows: goodness of fit for modelling studies; accuracy and precision of a method for methodological studies; and benchmarking and power analysis for comparative and investigative studies

**Table 9 materials-17-01698-t009:** Experimental studies distinguished evaluating the mechanical behaviour of the SB.

Reference	Type of Study	Pathology	Joint	Anatomical Position	Experimental Technique (Mode)	Dimensional Scale	Model	Apparent Elastic Modulus (GPa)	Tissue Elastic Modulus (GPa)	Tissue Hardness (GPa)	Condition
Ferguson et al. [129]	Co	OA	H	Femoral Head	Indentation	µm	EP	-	16.2–24.0	-	No reported cartilage damage
Ferguson et al. [129]	Co	OA	H	Femoral Head	Indentation	µm	EP	-	15.7–21.1	0.5–0.9 *	Severe cartilage damage
Peters et al. [16]	Co	OA; Aging	K	Femoral Condyle; Tibial Plateau	Indentation	µm	LE	-	12.56 ± 0.50 *	0.01–1.27	ICRS grade 0
Peters et al. [16]	Co	OA; Aging	K	Femoral Condyle; Tibial Plateau	Indentation	µm	LE	-	13.68 ± 0.60 *	0.01–1.27	ICRS grade 1
Peters et al. [16]	Co	OA; Aging	K	Femoral Condyle; Tibial Plateau	Indentation	µm	LE	-	14.05 ± 0.70 *	0.01–1.27	ICRS grade 2
Peters et al. [16]	Co	OA; Aging	K	Femoral Condyle; Tibial Plateau	Indentation	µm	LE	-	13.60 ± 1.00 *	0.01–1.27	ICRS grade 3
Peters et al. [16]	Co	OA; Aging	K	Femoral Condyle; Tibial Plateau	Indentation	µm	LE	-	17.20 ± 2.00 *	0.01–1.27	ICRS grade 4
Renault et al. [130]	In	OA	K	Tibial Plateau	Indentation	µm	LE	-	6.0–13.0 *	-	Light/severe cartilage damage

* = Data derived from graph. - = Data not reported. Type of study: In = Investigative. Co = Comparative. Pathology: OA = Osteoarthritis. Joint: H = Hip. K = Knee. Constitutive model: EP = Elasto-plastic. LE = Linear elastic.

## Data Availability

The data are reported in the tables of the manuscript.

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
