# Peer review of "Biomechanics of the Human Osteochondral Unit: A Systematic Review"

_materials, 2024, doi:10.3390/ma17071698_

Round 1

Reviewer 1 Report

Comments and Suggestions for Authors

The paper presents the current published studies on the osteochondral unti in a comprehensive review. I recommend this work for publication after the making the following alterations.

Here are some minor corrections prior to publication:

1) pg. 16 line 94; The analysis performed by MR is highly dependant on the resolution of the imaging. It is important to mention that any results (i.e. strain) found by any imaging technique and especially MR are affected  by the pixel/voxel size, especially when talking about tissue heterogeneity.

2) A very important point to consider, when presenting indentation results is the sample preparation. The different studies mentioned here all follow different experimental protocols that can give different results. Very few studies have attempted indentation on fresh, hydrated tissue (for example Davis S et al., 2023- on guinea pigs) and it should be made clear here that all these results are affected by the preparation and state of testing. 

3) DVC results also depends on the resolution of the subvolumes used in the analysis. Because of the complex multi-phasic system of OC it is important to mention that all these approaches and the results depend on the resolution. In the case of DVC for example if the subvolume is too big it does not account for the tissue heterogeneity. Whereas in FEA (although based on computational data and not experimental) it is easier to account for the heterogeneity in that sense.  

Comments on the Quality of English Language

1) The beginning of this sentence (pg.2 line 47) needs adjustment: "Despite there is not yet full understanding of the factors promoting or impairing the homeostasis  and, consequently, the mechanical response of the OC tissues, evidence suggests that changes in tissue strain levels may alter biochemical signals among tissues, as supported by mechano-regulation theories [7,8]."

2) pg2 line 52: "as osteoarthritis " should be "like osteoarthritis"

3) pg2 line 54: "therefore, producing changes of" should be therefore, produce changes on.

4) pg 2 line 58-61 please rewrite the sentence.

Reviewer 2 Report

Comments and Suggestions for Authors

The authors conducted an extensive literature survey providing a comprehensive review on the biomechanical properties of osteochondrial tissues in lower limb joints (hip, knee and ankle). An important outcome is the notion that the response of the OC as a whole has not yet been investigated. The review is of potential interest to many readers of materials.

Before acceptance, it is helpful to include an abbreviation list. Furthermore, as various methods are described, such as in Chapter 5, it is advisable to specifically mention in the final chapter "Conclusions and Future Perspectives" all the specific methods referred to by the authors.

Comments on the Quality of English Language

No language issues.

Reviewer 3 Report

Comments and Suggestions for Authors

The topic is very interesting for those people who are interested in biomechanics. The authors managed to summarise a huge amount of papers and to make a systematic review.

Reviewer 4 Report

Comments and Suggestions for Authors

For Title and Abstract:

1.     Title Redundancy: The title "Review: Biomechanics of the human osteochondral unit: A Systematic Review" contains redundancy. Consider simplifying it to "Biomechanics of the Human Osteochondral Unit: A Systematic Review" to maintain clarity and professionalism.

2.     Abstract Structure and Content:

·       The abstract could be structured more effectively by clearly delineating objectives, methodology, results, and conclusions.

·       The opening sentence is overly complex and could be simplified to enhance readability.

·       The statement about the importance of understanding biomechanics lacks specificity regarding the impact of this knowledge on clinical or research practices.

·       The methodology description lacks detail about the criteria for study selection and the systematic review process. It is crucial to specify how studies were evaluated for inclusion and the databases searched.

·       The phrase "Only studies on the human lower limb joints published between 2000 and 2022 were selected" could be integrated more smoothly into the methodology description.

·       The conclusion that "no study assessing the response of the entire osteochondral unit as a whole" is vague and lacks critical analysis or context regarding the implications of this finding.

·       The abstract hints at future advancements but does not adequately tie these to the gaps identified through the review. A more explicit connection between identified research gaps and future directions would strengthen the narrative.

3.     Keywords Consistency and Relevance:

·       The keywords are appropriate but could be expanded to include terms like "systematic review" and "biomechanical analysis" to enhance discoverability.

4.     Lack of Critical Analysis:

·       The text could benefit from a more critical analysis of the studies reviewed, including a discussion on the limitations of current research and how these limitations impact the understanding of osteochondral biomechanics.

·       The mention of methodological design preference is too brief. Expanding on why certain methodological designs were preferred and how they influence the reliability of findings would add depth to the review.

5.     Future Perspectives and Research Directions:

·       While future perspectives are mentioned, there is a lack of specific recommendations for future research directions based on the review findings. Detailing specific areas where further research is needed would be valuable.

·       The potential of imaging technologies and experimental techniques is mentioned vaguely. Elaborating on specific technologies and how they could overcome current limitations would make the argument more compelling.

6.     General Clarity and Coherence:

·       The abstract suffers from long sentences and complex phrasing, which could be simplified for better readability and comprehension.

·       Some statements are overly broad or lack specificity, undermining the academic rigor. Providing specific examples or evidence would enhance credibility.

7.     References to Methodological Rigor:

·       While the text mentions the implementation of a multi-criteria evaluation approach, it does not detail this approach or how it impacts the review's findings. Elaborating on the methodology would strengthen the review's credibility.

For Introduction:

1.     Clarity and Precision:

·       The introduction could benefit from a clearer and more concise definition of the OC unit at the outset to immediately ground readers unfamiliar with the term.

·       The sentence structure is occasionally convoluted, which may hinder comprehension. Shorter sentences and simpler construction could improve readability.

2.     Background Context:

·       While the introduction provides a basic overview of the OC unit and its importance, it lacks a detailed context of why this area is a critical subject of study. Incorporating statistics or specific examples of conditions affecting the OC unit could underscore the relevance of the research.

3.     Research Gap:

·       The introduction mentions the lack of full understanding and the importance of comprehensive knowledge but does not clearly articulate the specific gaps in current research. Explicitly stating these gaps could strengthen the rationale for the review.

4.     Methodological Overview:

·       The overview of the methods used in the systematic review is vague. Providing a brief but clear explanation of the search strategy, selection criteria, and types of studies included (e.g., in vivo, in vitro, computational) would offer readers a better understanding of the review's scope and limitations.

5.     Objective Clarity:

·       The objectives are listed in a manner that feels somewhat disjointed. Structuring this section with clear, bullet-pointed objectives or a structured paragraph that logically progresses from one aim to the next could enhance clarity.

6.     Preliminary Paragraph on Constitutive Models:

·       The mention of constitutive models and their historical context is informative but abruptly introduced. Integrating this discussion more smoothly with the rest of the introduction, perhaps by linking it directly to the identified research gaps, would improve flow.

7.     Use of Citations:

·       While citations are used, there's an opportunity to more effectively integrate them to support specific statements, particularly when referring to the importance of understanding the OC unit's biomechanics or discussing the impact of pathological changes.

8.     Transitions and Flow:

·       The transition between sections within the introduction (e.g., from the description of the OC unit to the discussion of homeostasis and mechanical crosstalk) could be smoother. Utilizing transitional phrases that clearly indicate shifts in focus or new sections of discussion would aid reader navigation.

9.     Technical Terms and Jargon:

·       The introduction uses specialized terms without always ensuring they are accessible to readers who might not be specialists in biomechanics. Where possible, brief definitions or explanations of key terms on their first occurrence could make the text more accessible.

10.  Future Perspectives:

·       While future perspectives are mentioned at the end, integrating a brief discussion on how addressing the research gaps identified might influence future research directions or clinical practices could offer a compelling closure to the introduction.

For Constitutive Models Of The Last Century:

1.     Introduction and Contextualization:

·       The section begins abruptly without adequately situating the reader in the context of why these models are important. A brief introduction highlighting the significance of understanding mechanical behavior through these models could set the stage more effectively.

2.     Technical Density and Accessibility:

·       The technical density of the text is high, which might limit accessibility for readers not deeply familiar with biomechanical modeling. Simplifying explanations or providing more context around complex terms could help.

3.     Chronological Organization:

·       The models are presented somewhat haphazardly. Organizing the models chronologically or by complexity could offer a clearer narrative flow, aiding reader comprehension.

4.     Comparative Analysis:

·       While various models are listed, there is a lack of comparative analysis discussing the advantages, limitations, or specific applications of each model. Adding a comparative dimension could significantly enrich the discussion.

5.     Citations and Referencing:

·       The section heavily relies on citations for detailed explanations of models, which is standard in academic writing. However, brief summaries of key findings or conclusions from these citations could make the text more informative and self-contained.

6.     Integration of Models with Experimental Data:

·       The text could benefit from examples or brief case studies demonstrating how these models have been applied in research, enhancing the practical relevance of the discussion.

7.     Addressing Model Limitations:

·       While some limitations and developments are mentioned, a more systematic discussion of the limitations of earlier models and how newer models seek to overcome these would provide valuable insights into the evolution of biomechanical modeling.

8.     Use of Visuals:

·       Incorporating figures or diagrams to visually represent the differences between models or to illustrate how they apply to the OC unit could significantly enhance understanding, especially for complex concepts.

9.     Clarity and Precision in Language:

·       The use of technical jargon is necessary but should be balanced with clear and precise language to ensure the text remains accessible to a broader academic audience.

10.  Future Directions and Emerging Models:

·       The section concludes with current models without discussing future directions or emerging trends in biomechanical modeling. Briefly mentioning ongoing research or theoretical advancements could provide a forward-looking perspective.

11.  Integration of Mechano-Electrochemical Phenomena:

·       While the inclusion of mechano-electrochemical phenomena is mentioned, elaborating on its significance in understanding tissue behavior could underscore the multidisciplinary nature of biomechanical research.

12.  Consideration for Damage Criteria:

·       The section intentionally neglects damage criteria; however, mentioning how these models might adapt or fail under conditions leading to tissue damage could provide a more holistic understanding of their applicability and limitations.

For Methods:

1.     Eligibility Criteria Specificity:

·       The eligibility criteria are broadly defined but could benefit from greater specificity. For instance, clarifying what constitutes "mechanical behaviour" or specifying types of experimental designs considered could reduce ambiguity.

2.     Search Strategies:

·       The search strategy is mentioned briefly; however, detailing the exact search terms used, the date range of the search, and any filters applied (e.g., language, publication type) would strengthen the reproducibility of the review.

3.     PRISMA Statement Adherence:

·       Mentioning adherence to the PRISMA statement is good practice, but including a PRISMA flow diagram in the appendix or main text would provide a visual summary of the study selection process, enhancing transparency and accountability.

4.     Study Selection Process:

·       The process for resolving disagreements between reviewers is mentioned, but the criteria or threshold for involving all authors in the decision-making process is unclear. Clarifying this would enhance understanding of the review's rigor.

5.     Data Extraction:

·       While data extraction is described, further details on the specific data extracted (e.g., types of mechanical properties, measurement techniques) and the rationale for the clustering approach would provide clarity on how the data synthesis supports the review's objectives.

6.     Systematic Assessment of Scientific Quality:

·       The adaptation of a specifically designed method for assessing scientific quality is innovative; however, a more detailed explanation of the Best-Worst Method (BWM) and its application in this context would be beneficial. Including a brief summary of the BWM in the main text, rather than solely in the appendix, could make the methodology more accessible to readers.

7.     Rating System and Aggregated Score:

·       The description of the rating system and the aggregation into an Aggregate Quality Score is concise but lacks detail on how exactly these scores are calculated and applied. Expanding on the scoring criteria and providing examples of how scores are assigned would add clarity.

8.     Use of Appendices:

·       While appendices are referenced for detailed methodologies and scoring systems, summarizing key elements of these appendices in the main text could improve the reader's ability to understand and evaluate the methodological approach without constantly referring to supplementary material.

9.     Language and Terminology:

·       The section is generally well-written but could benefit from careful proofreading to ensure consistency in terminology and clarity in descriptions. For example, ensuring consistent use of terms like "study" versus "article" or "paper" could improve readability.

10.  Clarity in Presentation of Results:

·       Mentioning that the findings of high-scoring studies will be tabulated is helpful, but outlining how these results will be discussed or analyzed in relation to the review's objectives would provide readers with a clearer understanding of the study's contribution to the field.

For Results and Discussion:

1.     Clarity and Structure:

·       The section could benefit from a clearer separation between results and discussion to enhance readability. Presenting the findings first, followed by a discussion that interprets these results in the context of existing literature, would improve the logical flow.

2.     Statistical Analysis:

·       The presentation of statistical analysis, such as the percentage of studies focusing on each tissue type and the aggregate quality score, is informative. However, it would be beneficial to include more detailed statistical analyses (e.g., statistical significance, confidence intervals) to support the claims made, especially when discussing trends over time or differences between tissues and joints.

3.     Figure References:

·       While the text references Figures 1, 2, and 3 to detail the study selection process, trends in research focus, and aggregate quality scores, a more detailed description or interpretation of these figures within the text would help readers who may not have immediate access to the figures understand the key findings.

4.     Methodological Quality Assessment:

·       The discussion on the methodological quality of studies, highlighted by the Aggregate Quality Score, is important. Expanding on how these scores were determined and discussing the implications of these quality assessments on the confidence readers can place in the reviewed findings would add depth to the analysis.

5.     Comparative Analysis:

·       The comparison between AC, SB, and TB focuses primarily on the quantity of research and identified trends. A deeper comparative analysis discussing why certain tissues have received more research attention and how the findings across tissues contribute to the overall understanding of OC biomechanics would be valuable.

6.     Trends and Gaps in Research:

·       While the text identifies trends in research focus, such as the increase in studies on AC and the knee joint, a discussion on the implications of these trends for future research and clinical practice is needed. Identifying gaps in the current literature based on the review findings could guide future studies.

7.     Discussion on Methodological Variability:

·       The section mentions different experimental techniques and models used to study AC's mechanical behavior but lacks a critical discussion on how methodological variability might affect the comparability of results across studies.

8.     Impact of Pathological Conditions:

·       The discussion on how pathological conditions like osteoarthritis (OA) affect the biomechanical properties of OC tissues is insightful. Further analysis on the clinical implications of these findings and suggestions for how this knowledge could inform treatment strategies would enhance the section's relevance.

9.     Technical Terms and Jargon:

·       The use of technical terms and jargon is appropriate for the academic audience, but providing brief explanations or definitions for complex concepts could make the section accessible to a broader readership, including clinicians and students.

10.  Future Directions:

·       While the section concludes with observations on the mechanical properties of OC tissues and their changes due to pathological conditions, explicitly stating future research directions based on these findings would be beneficial. Suggestions for addressing identified research gaps or exploring new models could inspire subsequent studies.

For Complementary Approaches To Investigate The OC Unit Biomechanics:

1.     Introduction and Context:

·       The introduction briefly mentions the importance of supporting mechanical assessments with other methods but lacks a clear statement on how these complementary approaches contribute to a holistic understanding of OC biomechanics. A more detailed rationale for integrating these methods would provide a stronger foundation for the section.

2.     Consistency and Detail in Descriptions:

·       Descriptions of various imaging and biochemical assessment techniques vary significantly in detail. Providing a consistent level of information for each method, including their specific applications, limitations, and how they complement mechanical testing, would improve the section's comprehensiveness.

3.     Critical Analysis of Techniques:

·       While the section lists various techniques, it lacks a critical analysis of their comparative advantages, limitations, and suitability for different research or clinical scenarios. Discussing these aspects would offer readers deeper insights into the selection of appropriate methods for specific investigative purposes.

4.     References to Figures or Tables:

·       The text does not reference any figures or tables that could visually summarize the information or provide examples of data obtained through these complementary approaches. Including such references, or suggesting their addition, could enhance understanding and engagement.

5.     Clarity on Specific Findings:

·       The section mentions that significant differences in tissue parameters have been found between different bone regions and under different physio-pathological conditions but does not provide specific examples or outcomes. Including key findings or trends observed in the literature could illustrate the practical implications of these complementary approaches.

6.     Discussion on Integration of Findings:

·       There is no discussion on how findings from these complementary approaches are integrated with mechanical assessments to provide a holistic view of OC biomechanics. Expanding on this aspect could demonstrate the multidisciplinary approach needed in this field.

7.     Consideration of Emerging Technologies:

·       The text could benefit from a mention of emerging technologies or recent advancements in imaging and biochemical assessment that may impact future research in OC biomechanics.

8.     Limitations and Future Directions:

·       While some limitations of specific techniques are mentioned, a broader discussion on the challenges of integrating data from diverse methodologies and potential solutions would be valuable. Additionally, suggesting future directions for developing or improving complementary approaches could inspire further research.

9.     Practical Applications and Clinical Relevance:

·       The section could be enhanced by discussing the clinical relevance of these complementary approaches, such as their role in diagnosing diseases, assessing treatment efficacy, or guiding surgical interventions.

10.  Accessibility and Jargon:

·       The use of technical jargon and complex descriptions may limit accessibility for readers not specialized in this area. Simplifying language where possible, or providing brief explanations of technical terms, could make the section more accessible to a broader audience.

For Conclusions and Future Perspectives:

1.     Comprehensiveness of Conclusions:

·       The conclusions briefly mention the focus on mechanical behaviour and the evaluation of single tissues within the OC unit but could be expanded to more comprehensively summarize the main findings, including significant achievements and limitations encountered in the reviewed studies. A more detailed synthesis of how these findings contribute to the current understanding of OC biomechanics would strengthen the section.

2.     Integration of Multidisciplinary Approaches:

·       Although the text acknowledges the multilayer structure of the OC unit, it could benefit from a more explicit discussion on the importance of integrating multidisciplinary approaches, including mechanical, biochemical, and imaging techniques, to fully understand the complex interactions within the OC unit.

3.     Specific Future Research Directions:

·       While mentioning the potential of full-field techniques for improving tissue engineering solutions, the section lacks specific recommendations for future research directions. Identifying particular areas where knowledge gaps exist, suggesting methodologies for addressing these gaps, and highlighting emerging technologies that could be leveraged would provide valuable guidance for future studies.

4.     Impact of Computational Advances:

·       The reference to the computing power of modern computers hints at the role of computational models in advancing OC biomechanics research. Elaborating on how computational tools and simulations could be used to predict the mechanical behaviour of the OC unit, assess treatment outcomes, or design better therapeutic strategies would underscore the importance of computational biomechanics in this field.

5.     Consideration of Pathologies:

·       The conclusions touch on pathologies impairing the OC unit but could further elaborate on how understanding the mechanical behaviour of OC tissues could lead to improved diagnostic and treatment options for specific conditions such as osteoarthritis. Discussing the potential for personalized medicine or targeted therapies based on biomechanical insights would highlight the clinical relevance of the research.

6.     Challenges and Limitations:

·       Acknowledging the challenges and limitations inherent in current research methodologies and how they might be overcome in future studies would provide a more balanced view. This could include technical limitations, the complexity of in vivo studies, or the need for standardization across research to enhance comparability.

7.     Engagement with the Broader Scientific Community:

·       Encouraging collaboration across disciplines and with the broader scientific community, including engineers, biologists, and clinicians, could be highlighted as a key strategy for advancing the field. Mentioning the importance of interdisciplinary research teams in tackling the complex problems associated with OC biomechanics would emphasize the need for collaborative efforts.

8.     Accessibility and Clarity:

·       Ensuring the conclusions are accessible to a broad audience, including non-specialists, by minimizing jargon and clearly explaining complex concepts would make the findings more impactful.

9.     Reflecting on Societal and Ethical Implications:

·       Briefly reflecting on the societal and ethical implications of advancements in OC tissue engineering, such as accessibility of new treatments and considerations for patient safety, could provide a well-rounded conclusion.

Comments on the Quality of English Language

Specific Comments and Suggestions:

1.     Sentence Structure:

·       Issue: Some sentences are excessively long and contain multiple clauses, making them difficult to follow.

·       Example & Suggestion: Look for sentences that exceed 25 words and consider breaking them into two or more sentences to improve readability.

2.     Technical Terminology:

·       Issue: Several technical terms are introduced without definitions, potentially alienating readers unfamiliar with the field.

·       Example & Suggestion: For the first instance of terms like "aggregate modulus" or "viscoelasticity," add a parenthetical definition or a brief explanatory sentence.

3.     Consistency in Terminology:

·       Issue: The manuscript alternates between different terms for similar concepts, such as "elastic behavior" and "elastic properties."

·       Example & Suggestion: Choose one term for each concept and apply it consistently throughout the document.

4.     Grammar and Syntax Errors:

·       Issue: Minor grammatical mistakes and awkward phrasing occasionally disrupt the flow.

·       Example & Suggestion: "The study of the comprehensive mechanical behaviour is possible" could be revised to "Studying the comprehensive mechanical behaviour is possible."

5.     Active vs. Passive Voice:

·       Issue: Overuse of passive voice may diminish the manuscript's engagement quality.

·       Example & Suggestion: "A full-field assessment was conducted" could become "We conducted a full-field assessment."

6.     Transitional Phrases:

·       Issue: The manuscript sometimes jumps between topics or sections without smooth transitions.

·       Example & Suggestion: Between major sections, add sentences like "Building on the previous analysis, we next explore..." to guide the reader.

7.     Punctuation and Formatting:

·       Issue: Inconsistent use of Oxford commas and formatting of lists can confuse readers.

·       Example & Suggestion: Ensure lists within sentences either consistently use or omit the Oxford comma.

Round 2

Reviewer 4 Report

Comments and Suggestions for Authors

Based on the detailed review and responses provided by the authors, it is evident that they have comprehensively addressed the feedback from me, as Reviewer 1, with diligence and academic rigor. The authors have not only refined the manuscript by integrating specific, constructive comments but also expanded upon critical discussions, elucidated future research directions, and enhanced the manuscript's accessibility through clearer language and supplementary materials. Such thorough engagement with the review process underlines the manuscript's contribution to the field of biomechanics, particularly regarding the osteochondral unit's mechanical properties and its relevance to pathologies like osteoarthritis.

The authors’ effort to bridge knowledge gaps through a multidisciplinary approach, incorporating mechanical, biochemical, and imaging techniques, signifies a valuable advancement towards understanding the complex interactions within the osteochondral unit. This holistic view is pivotal for developing diagnostic tools, evaluating treatment efficacy, and optimizing regenerative medicine strategies, thus holding significant promise for clinical applications.

In light of these comprehensive revisions and the importance of the study's findings, I strongly recommend that the manuscript be fully accepted for publication. The authors have effectively demonstrated that their work provides a significant contribution to the biomechanics community, offering insights that could guide future research and impact clinical practices. The meticulous attention to detail in responding to the review, coupled with the clarity and depth of the manuscript, ensures its suitability for publication, promising to enrich the academic discourse surrounding osteochondral biomechanics.